# Rational design of a multi-valent human papillomavirus vaccine by capsomere-hybrid co-assembly of virus-like particles

Daning Wang[1,2,3], Xinlin Liu[1,2,3], Minxi Wei[1,2,3], Ciying Qian[1,2], Shuo Song [1,2], Jie Chen[1,2], Zhiping Wang[1,2], Qin Xu[1,2], Yurou Yang[1,2], Maozhou He [1,2], Xin Chi[1,2], Shiwen Huang[1,2], Tingting Li[1,2], Zhibo Kong[1,2], Qingbing Zheng[1,2], Hai Yu[1,2], Yingbin Wang[1,2], Qinjian Zhao[1,2], Jun Zhang[1,2], Ningshao Xia [1,2✉], Ying Gu [1,2✉] & Shaowei Li [1,2✉]

The capsid of human papillomavirus (HPV) spontaneously arranges into a T = 7 icosahedral particle with 72 L1 pentameric capsomeres associating via disulfide bonds between Cys175 and Cys428. Here, we design a capsomere-hybrid virus-like particle (chVLP) to accommodate multiple types of L1 pentamers by the reciprocal assembly of single C175A and C428A L1 mutants, either of which alone encumbers L1 pentamer particle self-assembly. We show that co-assembly between any pair of C175A and C428A mutants across at least nine HPV genotypes occurs at a preferred equal molar stoichiometry, irrespective of the type or number of L1 sequences. A nine-valent chVLP vaccine—formed through the structural clustering of HPV epitopes—confers neutralization titers that are comparable with that of Gardasil 9 and elicits minor cross-neutralizing antibodies against some heterologous HPV types. These findings may pave the way for a new vaccine design that targets multiple pathogenic variants or cancer cells bearing diverse neoantigens.

[1] State Key Laboratory of Molecular Vaccinology and Molecular Diagnostics, School of Life Sciences, School of Public Health, Xiamen University, 361102 Xiamen, China. [2] National Institute of Diagnostics and Vaccine Development in Infectious Disease, Xiamen University, 361102 Xiamen, China. [3]These authors contributed equally: Daning Wang, Xinlin Liu, Minxi Wei. ✉email: nsxia@xmu.edu.cn; guying@xmu.edu.cn; shaowei@xmu.edu.cn

Human papillomavirus (HPV) is the major cause of cervical cancer and genital warts, which occur through the persistent infection of the proliferating cells of the epithelium. Cancers associated with HPV infection account for ~5% of all human cancers, predominantly of the cervix. HPV has evolved a strict infection mechanism, with specificity in terms of the type of species, tissue, and cell for infection. Moreover, the HPV virion is delivered into the host away from bodily fluids, resulting in a lower immune response by the host against the neutralization epitope of the capsid. In terms of this lower immune pressure and type-restricted neutralization mostly evoked in human antiviral immunity, the phylogenetics of HPV is complex, and there has been a gradual accumulation of more than 200 distinct genotypes[1].

HPV is a non-enveloped, double-stranded DNA virus with a $T = 7$ icosahedral capsid composed of 72 L1 pentamers (capsomeres) paired with L2 monomers[2]. L1 and L2 proteins have a comparable molecular weight (~55 kD), with minor diversity among the different HPV genotypes. The structures of the L1 protein in pentamer formation[3,4], $T = 1$ virus-like particles (VLPs)[5], and $T = 7$ icosahedral capsids[6] have been determined by X-ray crystallography and cryo-electron microscopy (cryo-EM) using proteins expressed in *E.coli*, yeast, insect cells, and mammalian cells. Unlike a regular icosahedral configuration, the HPV capsid is built only by pentameric blocks, with no hexamers. Indeed, structural analysis shows that the 72 L1 pentamers assemble into a dextra $T = 7$ icosahedral lattice by 5-ordinated (around a 5-fold axis) and 6-ordinated (at rest) interactions initiated by disulfide linkages between the Cys175 and Cys428 (as per HPV16 L1 numbering) of adjacent pentamers[6,7].

HPV type specificity is afforded by loop sequence diversity and structural differences, with most of the unique neutralization sites found on the five surface loops of the pentamer[4,8,9]. Although four prophylactic HPV vaccines—Gardasil 4[10], Gardasil 9[11], Cervarix[12], and Cecolin[13,14]—have been launched to market worldwide, they induce mostly type-restricted neutralizing antibodies and have limited cross-protection against a few of the non-vaccine types[15–18]. There are numerous efforts to design ways to tackle type specificity in a broad manner through the use of chimeric construct design[19–21], with some success achieved through neutralization epitope transplantation and epitope resurfacing to create a heterologous type L1 protein[22,23]. Pseudo-atomic resolution models for bovine[24] and human[6,25] papilloma viruses have highlighted the detailed interactions that occur among the 5-ordinated and 6-ordinated pentamers. However, few neutralization sites have been found at this pentamer–pentamer junction. In these models, loops comprising aa401-437 link the F-sheet and the C-terminal G-sheet, extending outward to form a suspended bridge; this structure was undetectable in all pentamer crystal structures[3–6]. Most pentamer pairs make contact through these suspended bridges, and are locked by a disulfide bond between Cys175 and Cys428[26–28]. Previous work has shown that the cysteine residues in this disulfide linkage play key roles during capsid assembly[28,29].

In our earlier study, we N-terminally truncated HPV L1 to improve its soluble expression in *E. coli* and assembly into VLPs in vitro[30]. Through maturation treatment and subsequent structural analysis of these VLPs, we identified the importance of the disulfide bond between C175 and C428[6]. Here, we sought to separately mutate residues C175 and C428 to alanine to block VLP self-assembly. We then combine L1 proteins bearing C175A or C428A mutations in solution, and found a recovery of particle formation through reciprocal assembly (i.e., C175A and C428A L1 proteins bind through their unmutated cysteine residues). Using stoichiometrical analysis, we find that optimal interactions occur with equal molar ratios of these mutant proteins, leading to

the formation of various capsomere-hybrid VLPs (chVLPs) when multiple HPV types are combined. The chVLPs resemble wild-type (WT) VLPs in morphology and physiochemical properties, but show different antigenicity and minor neutralization antibody response to heterologous HPVs. The strategy outlined in this work may shed light on ways to develop a pan-HPV vaccine and provides an appealing approach to simultaneously incorporate 72 antigenic determinants (one epitope per capsomere) into a single particle.

## Results

**Rational design for capsomere-hybrid HPV VLPs.** Disulfide linkages play key roles in the self-assembly of HPV L1 VLPs from pentameric capsomeres[27,28]. Studies show that Cys175 and Cys428 residues of HPV L1 proteins between adjacent pentamers combine to initiate the capsid assembly process when the both residues are present under oxidized and reduced conditions[28,29]. We surmised that a mutation in either residue would prohibit capsid formation. Using L1 proteins harboring either a Cys175Ala or a Cys428Ala (HPV16 numbering) mutation, we confirmed an inhibition of capsid formation, with the proteins folding as star-shaped pentamers that were incapable of self-assembling into HPV L1 VLPs. These so-called assembly-deficient pentamers showed abrogated Cys175-Cys428 disulfide bond formation within pentamer pairs (upper left, Fig. 1). From this, we speculated that the deficiency in self-assembly could be partly rescued by reciprocity: if we combined Cys175Ala and Cys428Ala mutants in solution, binding could be achieved between the unmutated cysteine residue in each mutant L1. In such a case, the assembled particles would retain only half as many disulfide bonds as compared with the wild-type (WT) (upper left cartoon, Fig. 1). To this end, we expressed and purified from *E. coli* C175A and C428A mutants of three HPV types: HPV6, HPV16, and HPV52. We found that HPVL1-C175A and -C428A mutants can pair-wise assemble for each of the three HPV types when combined (Fig. 1a, e, i). Furthermore, when we mixed different HPV types, heterologous type VLPs could be formed (Fig. 1b–d, f–h), which we hereafter refer to as capsomere-hybrid VLPs (chVLPs). These chVLPs resemble the WT L1 VLPs in size and morphology, as visualized in negative-staining transmission electron microscopy (TEM) (Fig. 1, Supplementary Fig. 1).

**Stoichiometry of capsomeres involved in hybrid-assembling HPV VLPs.** Our production of chVLPs inspired us to combine distinct types of HPV L1 capsomeres within the same VLP lattice. First, we determined the stoichiometry of chVLPs using serial ratio arrays (orthogonal combinations of 1, 1.5, 2, and 4 molar ratios) of C175A and C428A mutants. The assembled chVLPs and unassembled capsomeres were resolved by high-performance size-exclusion chromatography (HPSEC) and visualized using TEM (Fig. 2). Following various input concentrations of HPV16L1-C175A and HPV52L1-C428A, we found that only equal molar ratios led to the favorable assembly of chVLPs (Fig. 2a), with nearly no surplus of capsomeres in the HPSEC profile (Fig. 2b), and a trend that more unequal pairing (from 1:1.5, 1:2, and 1:4 of C175A:C428A mutants) led to poorer morphologies (Fig. 2a). In particular, no spherical particles could be observed in the 1:4 or 4:1 ratio, which suggested that chVLPs comprise equal numbers of C175A and C428A mutant pentamers. In addition, unpaired pentamers in other ratios disturbed the assembly of chVLPs through irregular disulfide associations, with most surplus pentamers free from assembly (Fig. 2b).

The chVLP fractions (~11.5 min retention time) in the HPSEC profiles of HPV16L1-C175A–HPV52L1-C428A and HPV52L1-C175A–HPV16L1-C428A were harvested and subjected to

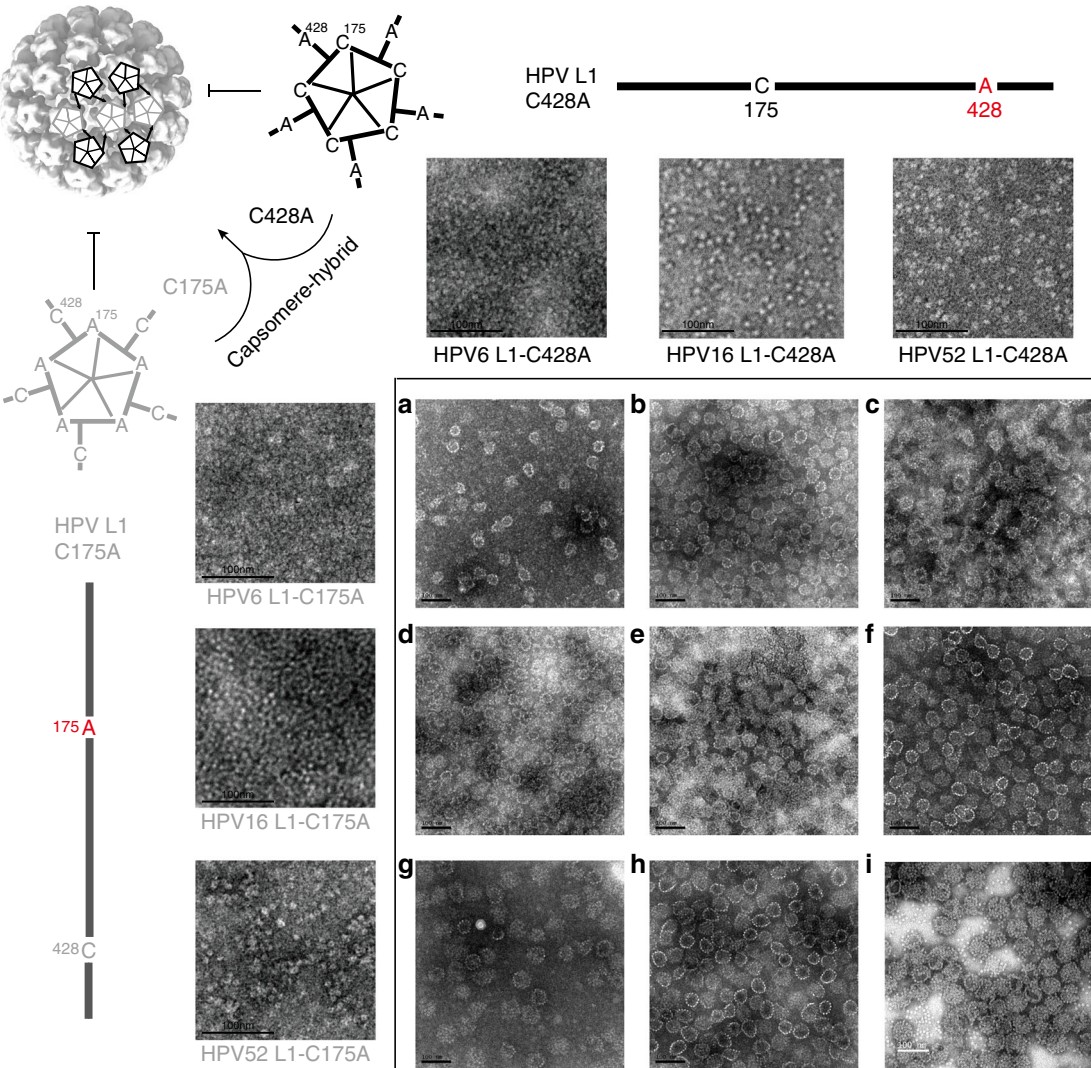

**Fig. 1 Schematic presentation and negative-staining transmission electron microscopy (TEM) of assembly-deficient HPV L1 pentamers and their capsomere-hybrid assembly via the formation of single inter-disulfide bonds.** Scale bar, 100 nm. The replacement of Cys175 or Cys428 of HPV L1 with Ala impedes self-assembly into VLPs; albeit, folding to capsomere-characteristic donut-like pentamers is retained, as shown in the TEM views. Left/top panel: mutants of HPV6, HPV16, and HPV52 L1-C175A/C428A were de-thiolated on aa175/aa428 but preserve the thiol group on aa428/aa175, which abrogate their self-assembly into VLPs with respect to the WT HPV L1. **a–i** Right lower panel: combination of any C175 mutant and any C428 mutant (homologous type or heterologous type of HPV), allows chVLP assembly of two single-point mutated pentamers with reciprocal disulfide bonding potentials. Schematic cartoon (left top region) illustrates the reciprocal linkage of Cys175 and Cys428 from pentamer mutants C428A and C175A, respectively, for chVLP assembly. Also refer to Supplementary Figs. 1 and 9. One representative image from three biological repeats is shown.

SDS-PAGE, western blotting, and double-antibody sandwich enzyme-linked immunosorbent assay (ELISA) with type-specific monoclonal antibodies (mAbs). Consistent with our HPSEC profiles for equal molar ratios, chVLPs, confirmed to simultaneously harbor C175A and C428A pentamers (Supplementary Fig. 2), resolved with comparable amounts of HPV16 (502 aa) and HPV52 (490 aa) L1 proteins in the SDS-PAGE gel, and with approximately equal protein amounts noted with HPV16- and HPV52-specific mAbs in western blotting (Fig. 2c).

**Characterization of HPV chVLPs.** Next, we further characterized the HPV chVLPs in terms of their particulate, thermal stability, and structural characteristics. Through particle-based analysis, we found that HPV16L1-C175A–HPV52L1-C428A chVLPs and the reciprocal VLPs (HPV52L1-C175A–HPV16L1-C428A) shared comparable retention times (~12 min), and both showed a single-component distribution in the HPSEC profiles

(Fig. 3a), with comparable sedimentation coefficients of 123.4S and 117.4S in the analytical ultra-centrifugation (AUC) profiles (Fig. 3b). In addition, these reciprocal chVLPs had comparable hydrated diameters of 62.1 nm and 69.6 nm, respectively, in the dynamic light scattering (DLS) analysis (Fig. 3c), which were very similar to those for WT HPV52 VLPs. It should be noted that HPV16 VLPs were smaller in size and had a lower S-value than HPV52 VLPs or HPV16/52 chVLPs; this is due to the varied N-terminal truncations of HPV16 and HPV52 involved in particle assembly[30]. In the differential scanning calorimetry (DSC) curves, HPV16 VLPs and HPV16L1-C175A pentamers possessed higher melting temperatures (Tm) ~5–10 °C over that of HPV52: 68.9 °C vs. 64.2 °C, respectively, for VLPs, and 66.9 °C vs. 57.0 °C, respectively, for pentamers. On the other hand, HPV16/52 chVLPs exhibited two Tm peaks: one major peak was located at ~67 °C, which was between the values for HPV16 and HPV52 VLPs, and a minor peak at ~60 °C, between the values of C175A and C428A pentamers (Fig. 3d). These results indicate the

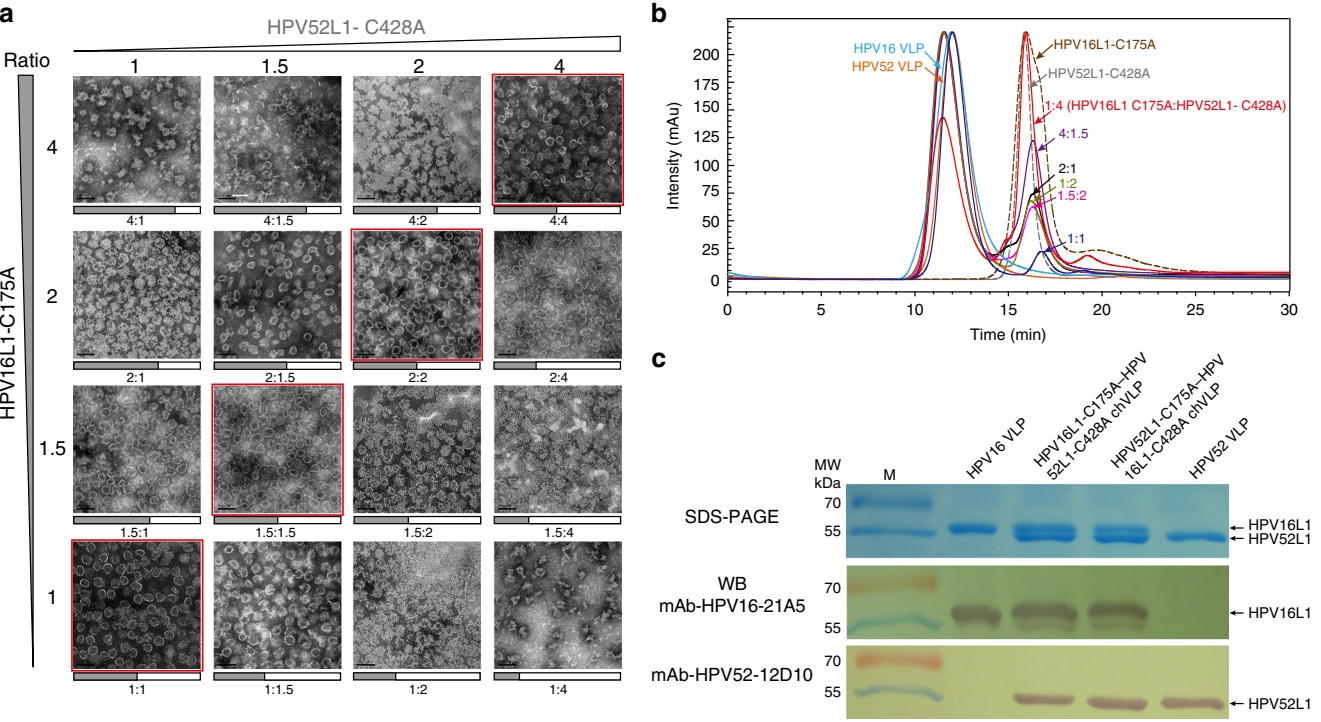

**Fig. 2 Stoichiometry optimization of capsomere-hybrid assembly of double-type HPV VLPs. a** Transmission electron microscopy (TEM) views show the ratio profiles of various molar pairings of HPV16L1-C175A and HPV52L1-C428A, with better VLP assembly in the pictures along the forward diagonal of the array (see images with red boxes). This suggests a preference for an equal molar ratio of two distinct capsomeres in hybrid assembly. Scale bar, 100 nm. One representative image from three biological repeats is shown. **b** Size-exclusion chromatography resolves the components of hybrid-assembling samples with various molar ratio mixing. The resultant chVLPs were retained at earlier retention time with respect to original pentamer. The unassembled pentamer component nearly disappears in the elution curve of 1:1(molar ratio) mixture of HPV16L1-C175A and HPV52L1-C428A. HPV16 and HPV52 VLPs, HPV16L1-C175A and HPV52L1-C428A mutants were used as elution markers in the chromatography. **c** Coomassie-stained SDS-PAGE showed the comparable amounts of HPV16 and HPV52 L1 mutants within assembled HPV16L1-C175A–HPV52L1-C428A chVLPs and HPV52L1-C175A–HPV16L1-C428A chVLPs. The different length of HPV16 and HPV52 L1 constructs allows to be separated and quantified in gel and they were verified by type-specific mAb western-blotting, respectively. The results reveal the optimized stoichiometry for hybrid-assembling is equal molar ratio of paired pentamer mutants, originals, and overloading of either pentamer over 1:1 ratio disturbs the assembly procedure and leads to irregular aggregation. The uncropped original scans can be found in Supplementary Fig. 18.

distinctive innate thermal stabilities of the chVLPs as compared with the WT VLPs, and may reflect that chVLPs are constituted by two components and a reduced number of disulfide bonds.

We therefore next tested disulfide bond abundancies through a quantification procedure, as previously described by Yongsawat-digul and Park[31,32]. Disulfide bond abundancies were measured as $8.8 \pm 0.4$, $16.1 \pm 1.9$, and $15.6 \pm 0.6\,\mu mol/g$ for HPV16L1-C175A–HPV52L1-C428A chVLPs, HPV16 VLP, and HPV52 VLP, respectively (Supplementary Fig. 3), confirming that half as many disulfide bonds formed in the chVLPs compared with the WT VLPs. Despite of ½ disulfide bonds formed, the chVLPs exhibited comparable stability with that of WT VLPs during storage at 4 °C, 25 °C, and 37 °C for up to 10 weeks, as assessed via protein integrity with SDS-PAGE and western blotting (Supplementary Fig. 4), thermal stability with DSC (Supplementary Fig. 5), sedimentation coefficient values with AUC (Supplementary Fig. 6), particle component retention time with HPSEC (Supplementary Fig. 7), hydrodynamic radius with DLS (Supplementary Fig. 8) and VLP morphology under TEM (Supplementary Fig. 9).

The cryo-EM structure of the HPV16L1-C175A–HPV52L1-C428A chVLP was determined at 26.1-Å resolution (Supplementary Table 1) and compared with known structures of WT HPV16 and HPV52 VLPs (EMDB no. 6795 and 6919)[30,33]. chVLPs assumed a dextra $T = 7$ icosahedral arrangement, similar to the WT VLP but with a diameter of 55.0 nm, which more

closely resembles that of HPV52; these results are consistent with those of the aforementioned DLS and AUC analyses (Fig. 3e). However, the particle structures of these two types of capsomeres ascribed to HPV16L1-C175A and HPV52L1-C428A pentamers were indistinguishable at this lower resolution, and thus we could not identify their assembling scenarios and distributions within the chVLPs.

**Hybrid-assembly of pseudoviruses in 293FT cells**. To examine whether the hybrid-assembly of VLPs that form in vitro can be applied for HPV pseudoviruses (PsVs) generation in vivo, we introduced L1-C175A and L1-C428A mutations into the L1 gene and transfected this gene into 293FT cells, together with the L2 and GFP reporter genes for PsVs production, as described by Schiller[34]. To assure favorable co-assembly, the expression levels of HPV16L1-C175A and HPV52L1-C428A were quantified by quantitative ELISA (Supplementary Fig. 10) and optimized to comparable levels according to the input ratio of the transfected L1 plasmids. We then used negative-staining TEM to examine the morphology of the resultant PsVs, with the numbers of fluoro-spots counted to measure infectivity using an ImmunoSpot reader. Similar to the results of the hybrid-assembly of chVLPs in vitro, individual HPV16L1-C175A or HPV52L1-C428A proteins could not assemble into particles in 293FT cells (Fig. 4a) but could do so when co-expressed with the complementary mutated

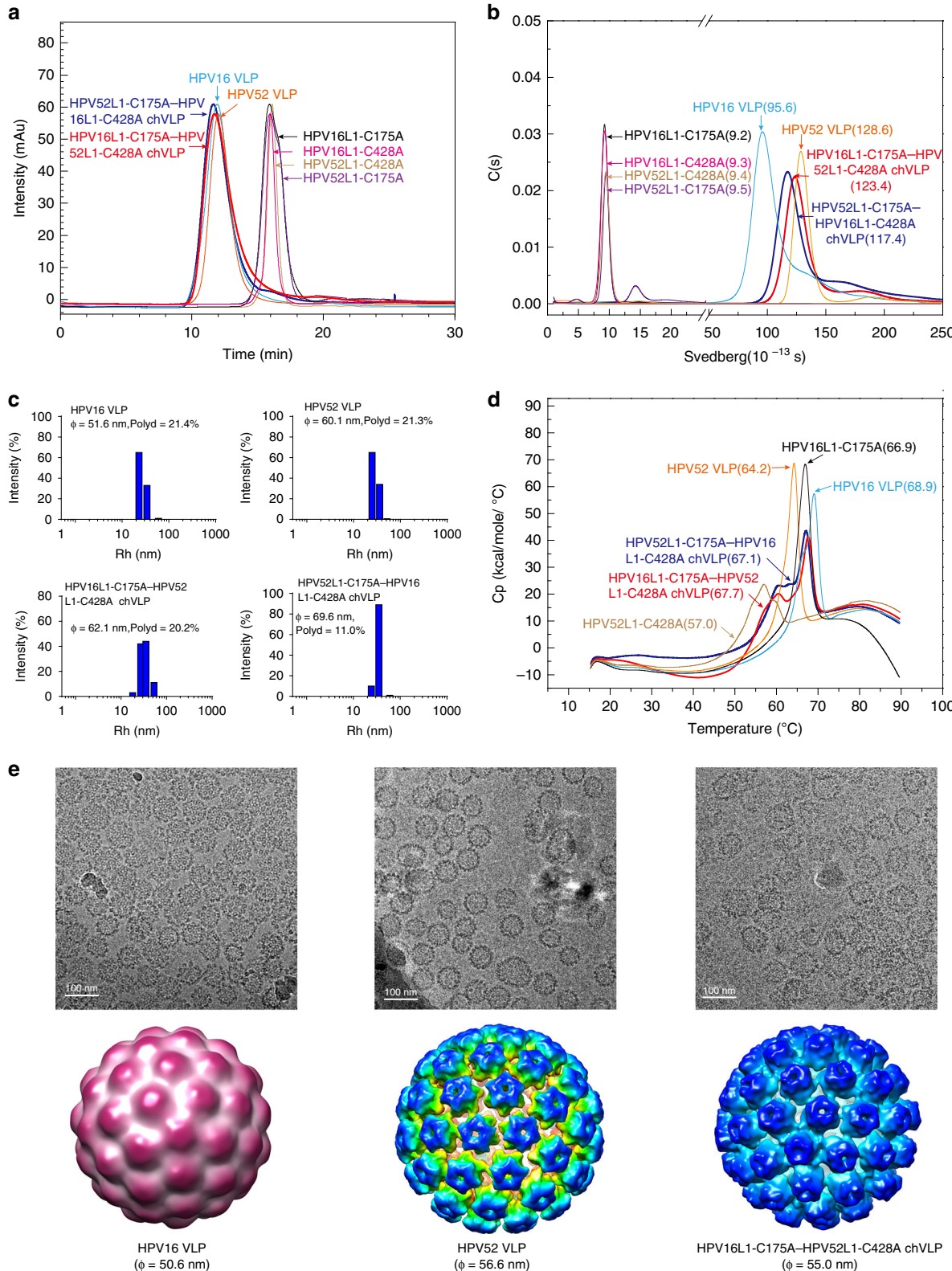

L1 protein. Interestingly, with the assistance of the L2 proteins same genotype as L1, either HPV16L1-C175A or HPV52L1-C428A alone could assemble into particles; albeit, with poor morphology in TEM. This is consistent with the results reported by Ishii[35] on the assistance provided by the L2 protein during assembly in the presence of assembly-deficient L1. It is note-worthy that these HPV16L1-C175A only and HPV52L1-C428A

only particles assisted by L2 lost the ability to infect 239FT cells (Fig. 4b). Like WT HPV16 or HPV52 PsVs, the co-assembly of the HPV16L1-C175A and HPV52L1-C428A with the HPV16L2 or HPV52L2 (or both L2 proteins) led to good particle formation in the TEM and conferred marked infectivity in the fluorospot assay (Fig. 4c). Taken together, the pairing of L1-C175A and L1-C428A mutants with L2 involvement leads to co-assembly in

**Fig. 3 Physicochemical properties and structural characterization of chVLPs. a** High-performance size-exclusion chromatography (HPSEC) profiles and **b** sedimentation velocity profiles of HPV16 VLPs, HPV52 VLPs, HPV16L1-C175A–HPV52L1-C428A chVLPs, HPV52L1-C175A–HPV16L1-C428A chVLPs, HPV16L1-C175A, HPV52L1-C175A, HPV16L1-C428A, and HPV52L1-C428A. **c** Dynamic light scattering analysis of HPV16 VLPs, HPV52 VLPs, HPV16L1-C175A–HPV52L1-C428A chVLPs, and HPV52L1-C175A–HPV16L1-C428A chVLPs. **d** Differential scanning calorimetry reveals an additional melted phase of the original capsomeres for HPV16L1-C175A–HPV52L1-C428A chVLPs and HPV52L1-C175A–HPV16L1-C428A chVLPs as compared with WT VLPs. The dominant phase transition peak lies between the parental HPV16 and HPV52 VLPs, which have a single Tm peak in the scanning curves. Tm values for HPV16 VLPs, HPV52 VLPs, HPV16L1-C175A–HPV52L1-C428A chVLPs, HPV52L1-C175A–HPV16L1-C428A chVLPs, and HPV16-C175A and HPV52-C428A L1 proteins were 68.9 °C, 64.2 °C, 67.7 °C, 67.1 °C, 66.9 °C, and 57.0 °C, respectively. **e** Micrographs of vitrified HPV16, 52 L1 VLPs and HPV16L1-C175A–HPV52L1-C428A chVLPs (top). Scale bar, 100 nm. One representative image from three biological repeats is shown. Reconstructed 3D cryo-electron microscopy (cryo-EM) maps of HPV16 and HPV52 L1 VLPs and HPV16L1-C175A–HPV52L1-C428A chVLPs (below). In terms of morphology, chVLPs present as a $T = 7$ icosahedral arrangement similar to the wild-type VLPs.

293FT cells and rates of infectivity similar to that of WT HPV PsVs.

**Co-assembly of HPV VLPs with more than two types of pentamers**. We next endeavored to assemble a single chVLP with more than two types of HPV L1 proteins bearing either the C175A or C428A mutations. Seven N-terminally truncated L1 proteins from HPV6, −16, −33, −45, −52, −58, and −59 were expressed in *E.coli*[6,30,33,36,37]. For capsomere-hybrid creation, we first created five tri-type permutations: HPV45/59L1-C175A–HPV52L1-C428A, HPV16/33L1-C175A–HPV58L1-C428A, HPV16/59L1-C175A–HPV52L1-C428A, HPV33L1-C175A–HPV16/58L1-C428A, and HPV45L1-C175A–HPV16/52L1-C428A. These three mutants were subjected to co-assembly, combining one or two types of C175A mixed with one or two types of C428A at equal molar ratios of C175A and C428A mutations. These resultant chVLPs exhibited similar sizes and morphology (Fig. 5a–e) as that of di-type co-assembled chVLPs (Fig. 1) in negative-staining TEM. Intriguingly, this strategy could be further extrapolated to the hybrid assembly of combinations bearing 4, 5, 6, and 7 types of HPV L1s that were rationed to have equal molar concentrations of C175A and C428A mutations (Fig. 5f–p).

Double-antibody sandwich ELISA based on type-specific mAbs (Supplementary Fig. 11) was used to assess whether more than one type of L1-C175A or L1-C428A pentamer coexisted in the chVLPs. A mixture of the L1-C175A mutants or L1-C428A mutants of various genotypes or a mixture of the corresponding WT VLPs served as controls. The reactive curves in the plots of OD$_{450}$ vs. protein concentration indicated that HPV45L1- and HPV59L1-C175A co-assembled in a tri-type chVLP (with HPV52L1-C428A; Supplementary Fig. 12a), HPV16L1- and HPV58L1-C428A also in a tri-type (with HPV33L1-C175A; Supplementary Fig. 12b), HPV16L1- and HPV52L1-C428A as a tetra-type (with HPV45/59L1-C175A; Supplementary Fig. 12c), and HPV16L1- and HPV59L1-C175A as a penta-type (along with HPV45L1-C175A and HPV52/58L1-C428A; Supplementary Fig. 12d). There was no detectable signal for the control samples. Our findings suggest that the reciprocal assembly of a chVLP by equal-molar pairing of C175A and C428A capsomers is independent of the specific L1 sequences and could theoretically contain up to 72 types of pentamers according to $T = 7$ icosahedral arrangement.

**Antigenicity variation on chVLPs by hybrid-assembly**. HPV evolves into considerable genotypes with type-specific neutralization sites that equip HPV with a means to avoid host immune surveillance[38,39]. It is well-known that type-specific neutralization epitopes are located on the surface loops of intra-pentamers[4] and inter-pentamers[40]. Therefore, we next investigated the potential antigenic variation of hybrid-assembled VLPs

using HPV16/52 type-specific mAbs as well as mAbs raised from HPV16/52 chVLPs. The reactivities of both HPV16L1-C175A–HPV52L1-C428A and HPV52L1-C175A–HPV16L1-C428A chVLPs were measured against three panels of mAbs using median effective concentration (EC$_{50}$) calculations through ELISA; WT HPV16/HPV52 VLPs, HPV16L1-C175A/C428A pentamers, HPV52L1-C175A/C428A pentamers, HPV16L1-C175A–HPV16L1-C428A VLPs, and HPV52L1-C175A–HPV52L1-C428A VLPs served as controls. Of note, the rEC$_{50}$ was calculated as the reciprocal of the EC$_{50}$ value in tests against the EC$_{50}$ of WT VLPs to reflect the variations in hybrid-assembly antigenicity. Intriguingly, we show that the two reciprocal forms of HPV16/52 chVLPs present different rEC$_{50}$ profiles against the anti-HPV16 or HPV52 mAb panel, where the reactivity of HPV16L1-C175A–HPV52L1-C428A chVLPs is more susceptible to antibody binding upon hybrid-assembly than is the reciprocal chVLPs; this is shown by the greater decline in the reactivities against some of the anti-HPV16 and anti-HPV52 mAbs. As expected, most of the reactivities against mAbs recognizing linear epitopes remained unchanged. Of note, the well-characterized immunodominant neutralization epitope identified by mAb HPV16.V5[41] is maintained in the two forms of chVLPs. In contrast, hybrid-assembled VLPs of either homo-HPV16 or -HPV52 shared similar overall reactivity profiles with that of WT HPV16 or HPV52, respectively (Fig. 6a, b).

Next, we raised mAbs through the immunization of mice with HPV16L1-C175A–HPV52L1-C428A chVLPs, and found that mAb 10C3 specifically targets chVLPs. Moreover, the hybrid-assembled HPV16/52 VLPs exhibited cross-type reactivity with heterologous type mAbs, such as HPV33, −45, −58, and −59, as compared with WT VLPs (Fig. 6c). Taken together, the HPV16/52 hybrid assembly maintains most of the type-specific epitopes of both genotypes; albeit, some epitopes exhibit perturbations, and this resurfacing creates new epitopes as compared with the WT ones, some of which cross-react with mAbs specific to heterologous types, such as HPV33, −45, −58, and −59.

**Immunogenicity of hybrid-assembling HPV VLPs**. In light of the cross hetero-genotype antigenicity created after hybrid-assembly, we next evaluated the immunogenicity of various chVLPs and their ability to elicit potential cross hetero-genotype neutralization antibodies in mice. First, mice were administered with di-type chVLPs (HPV16L1-C175A–HPV52L1-C428A) at 0.002, 0.008, 0.04, 0.2, 1.0, and 5.0 μg dosages, with a mixture of HPV16 and HPV52 VLPs at the same dosage serving as a control. The neutralization assay for both HPV16 and HPV52 showed that chVLPs possessed comparably high immunogenicity as compared with that of the mixture of VLPs in regards to both neutralization titer elicitation and dosage-dependent response. Indeed, as low as 0.008 μg could still produce about a 3-log HPV16 or HPV 52 neutralization titer (Fig. 7a, b), similar to that

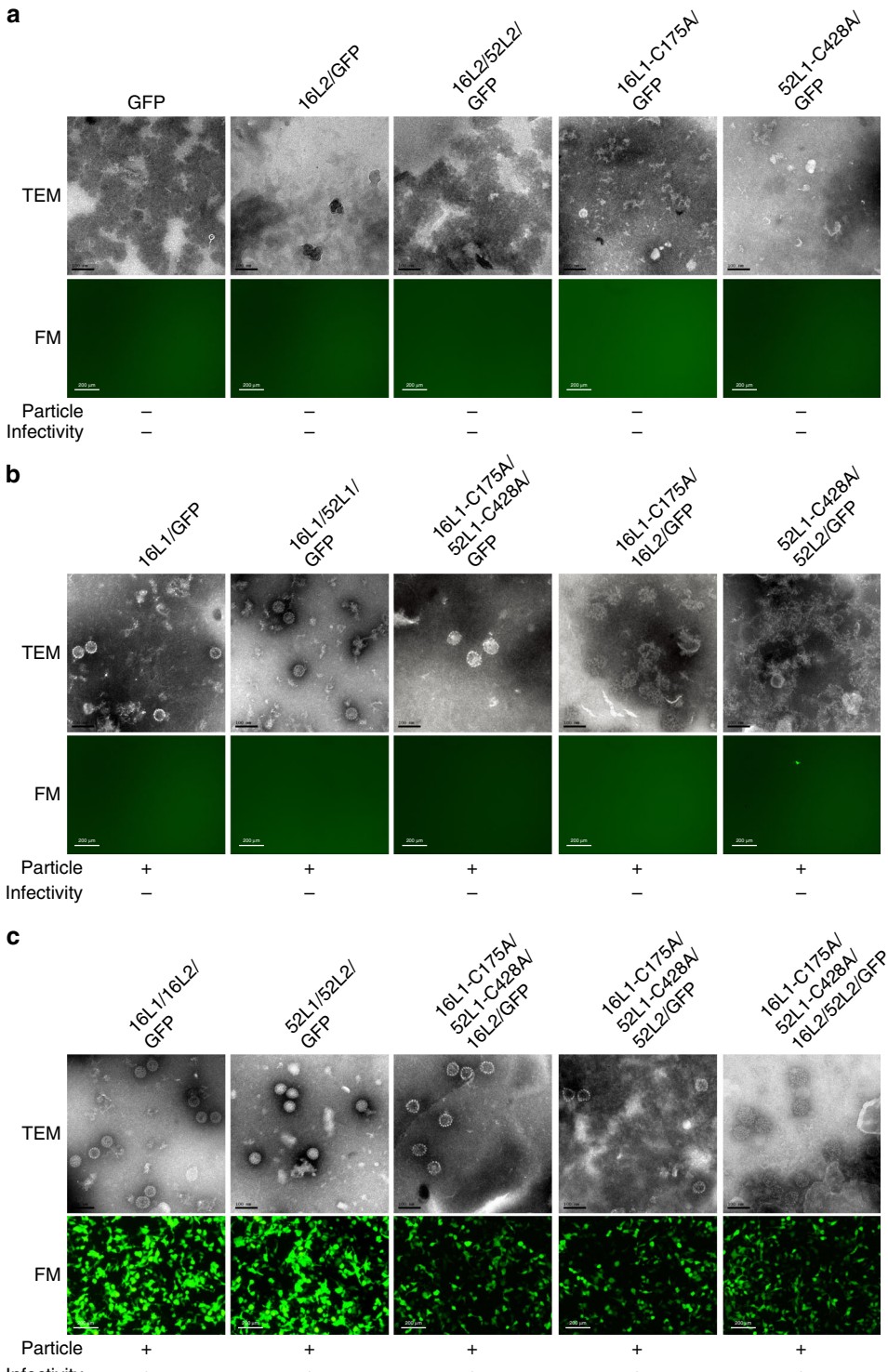

**Fig. 4 Hybrid-assembly of HPV16 and HPV52 PsVs in 293FT cells.** In each experiment, the plasmid N31-EGFP gene was co-transfected with plasmid L1 and L2 into 293FT cells to be packed into possible particles. GFP was used to detect PsVs infectivity with fluorescence microscopy (FM). The particles were analyzed by negative-staining transmission electron microscopy (TEM). **a** HPV16L1-C175A or HPV52L1-C428A constructs lost the capacity for particle self-assembly and infectivity. **b** HPV16L1-C175A and HPV52L1-C428A can hybrid-assemble into good particles similar to that of wild-type HPV16L1 and HPV52L1. HPV16L1-C175A or HPV52L1-C428A alone can assemble into particles with the assistance of the corresponding L2; although, the particles did not appear to be well-formed in TEM view. These particles lost their ability to infect 239FT cells. **c** HPV16L1-C175A and HPV52L1-C428A hybrid-assemble into infective PsVs with involvement of HPV16 or HPV52 L2 or both. Scale bar in TEM view, 100 nm; Scale bar in FM view, 200 µm. One representative image from three biological repeats is shown for each group.

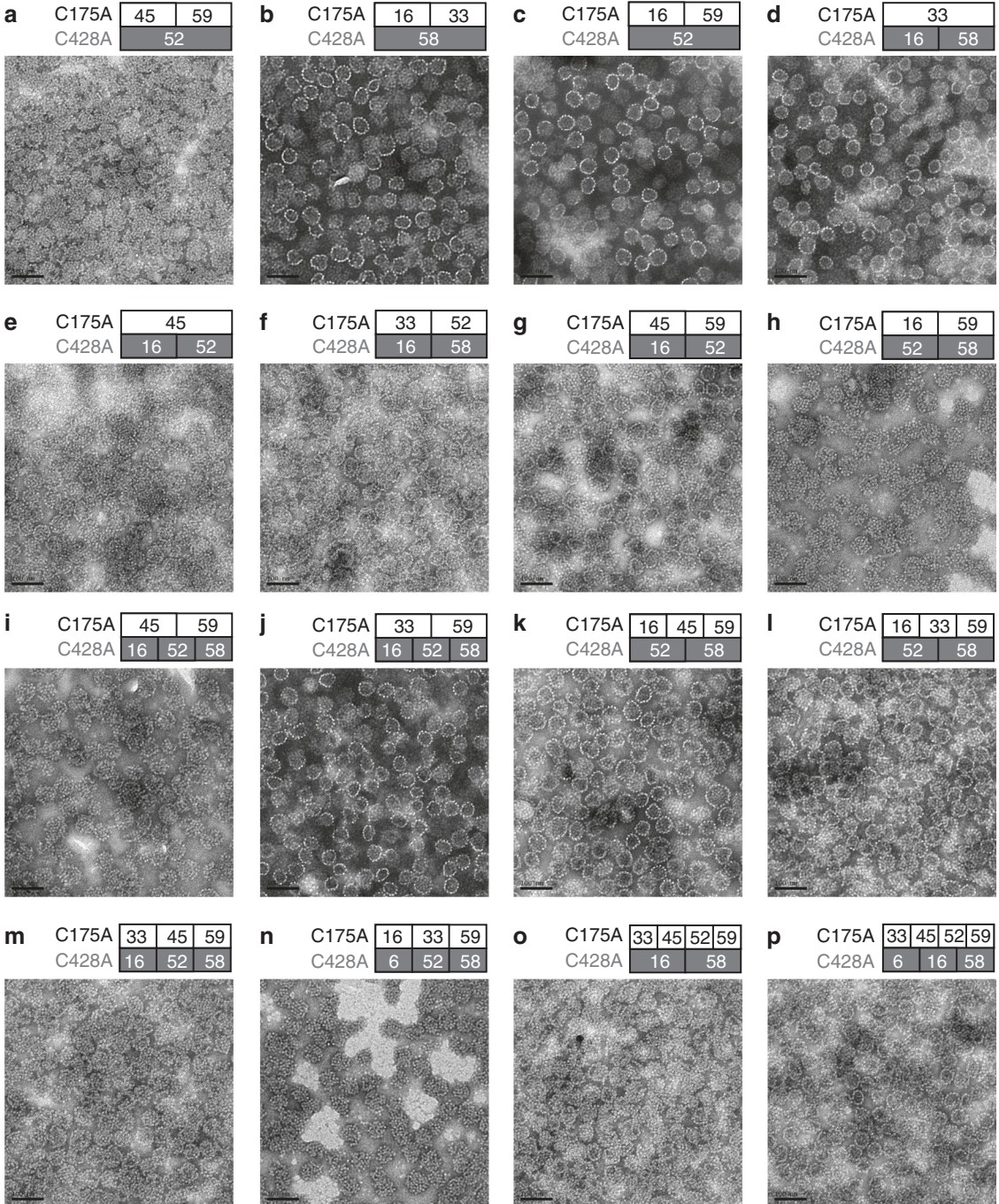

**Fig. 5 Hybrid-assembly of more than two types of HPV capsomeres. a–e** Three types of capsomeres are involved in the assembly, with a 1:1 molar ratio of the total C175A mutants and the total C428A mutants. **f–h** Tetra-type hybrid-assembly. **i–l** Penta-type hybrid-assembly. **m–o** Hexa-type hybrid-assembly. **p** Hepta-type hybrid-assembly. Scale bar, 100 nm. One representative image from three biological repeats is shown for each group.

of the half-effective dosage (ED$_{50}$) of HPV16 in the HPV16/18 bivalent vaccine in previous work[33].

Second, mice antisera produced with the high dosage (5 μg) were further analyzed with the neutralization of heterologous genotype HPV6, −11, −18, −31, −33, −45, −58, and −59 in addition to homologous genotypes HPV16 and −52. As expected, the mixture of VLPs could not induce cross-genotype neutralization, whereas the chVLPs (HPV16L1-C175A–HPV52L1-C428A) significantly elicited a 1–2-log higher neutralization titer for HPV33 and −58 (Fig. 7c); this is similar to the antigenicity profile of chVLPs (Fig. 6c) and associated with the phylogenetics of these HPV types (Supplementary Fig. 13).

Next, mice were immunized with other multi-type chVLPs (more than two types) at 2.5–5.0 μg (Supplementary Tables 2 and 3). We then measured immune sera against 10 genotypes using a neutralization assay. We found that all chVLPs could elicit heterologous genotype neutralization titers; albeit, with variable profiles up to a 1-log increase. Overall, we show that tri-type chVLPs could cover 5- (Fig. 7d, e) to 6-type (Fig. 7f) neutralization; tetra-type chVLPs could cover a maximum of 6 types (Fig. 7h); the penta-type and hexa-type chVLPs could also cover up to 8 types (Fig. 7j, k); and the hepta-type chVLPs could confer neutralization against all 10 types of HPV (Fig. 7l).

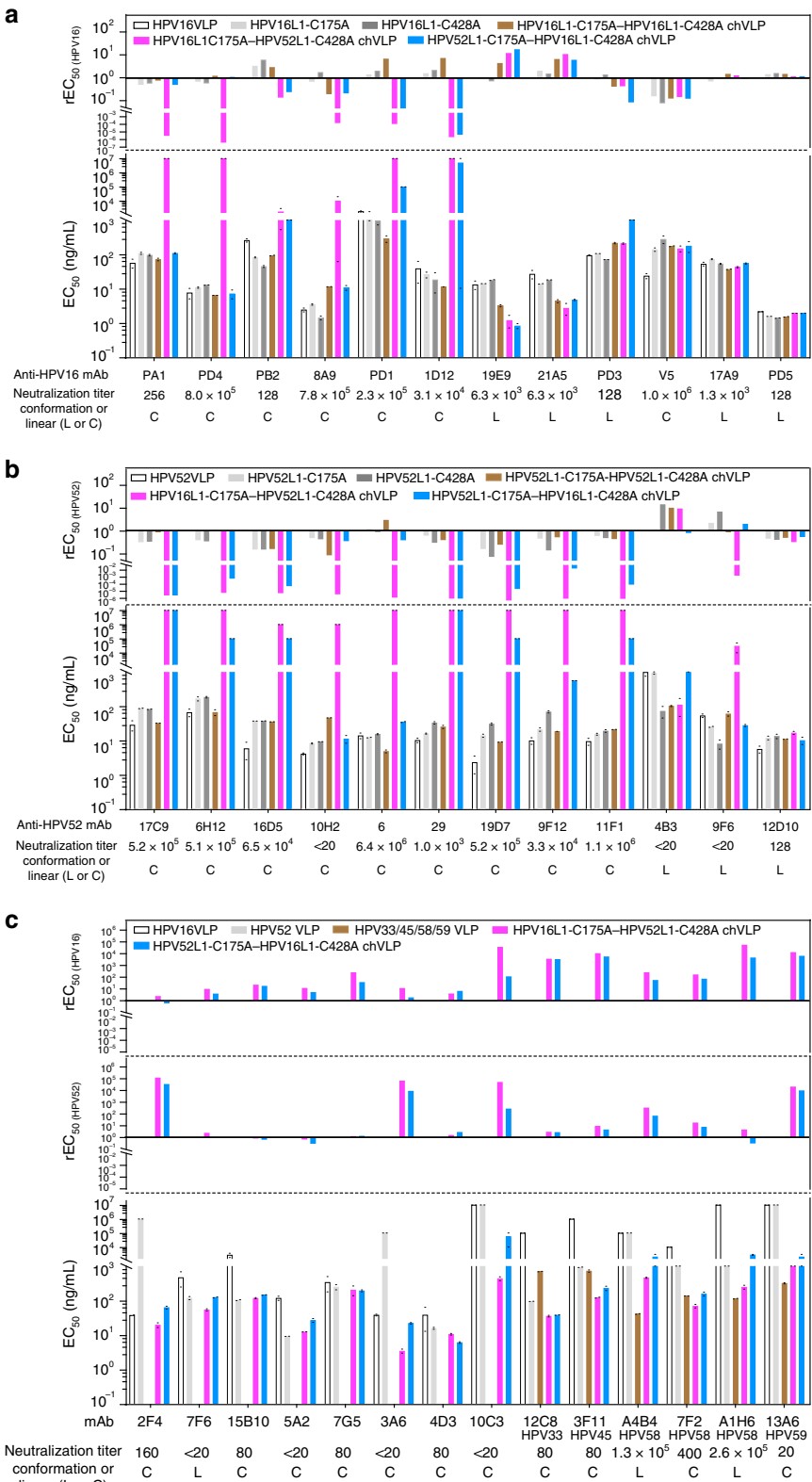

**Fig. 6 Varied antigenicity of chVLPs. a** HPV16-specific mAbs, **b** HPV52-specific mAbs, and **c** mAbs raised from HPV16L1-C175A–HPV52L1-C428A chVLPs were used to probe epitope variation. The reactivities of VLPs or pentamer mutants against various mAbs were quantified as $EC_{50}$ values (bottom panels), and their corresponding $rEC_{50}$ values (defined as $1/(EC_{50}$ [chVLPs, pentamers or other VLPs]$/EC_{50}$ [HPV16 or HPV52 VLPs]) were shown in the upper panels. A larger $rEC_{50}$ value indicates a higher reactivity for chVLPs against mAbs in test with respect to WT VLPs. The $EC_{50}$ value was obtained from the mean value of two repeated data. Source data are provided as a source data file.

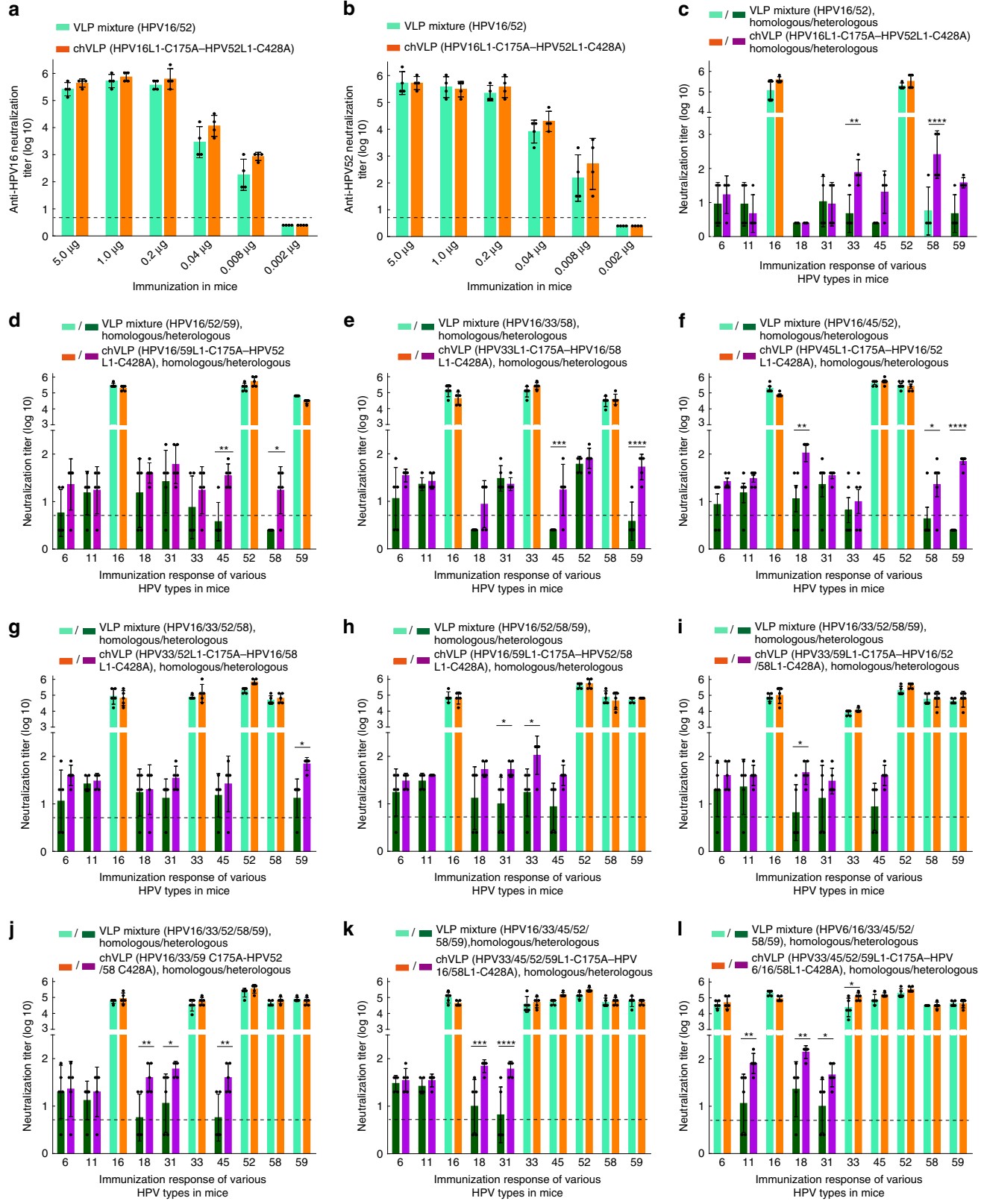

Finally, we generated a nona-type chVLP according to the composition and dosage of L1s in the Gardasil 9 formulation, and measured the neutralizing antibody titers with 20 types of HPV PsVs—nine homologous types (HPV 6, −11, −16, −18, −31, −33, −45, −52, −58) and 11 heterologous ones (HPV 26, −35, −39, −51, −53, −56, −59, −66, −68, −69, −70). Consistent with other multi-type chVLPs (Figs. 5 and 7), the nona-type chVLP showed good physiochemical and particle nature (Supplementary Fig. 14) and elicited a high neutralizing antibody response against the nine homologous types of HPV similar to that of the WT VLP mixture and Gardasil 9, moreover, additionally induced some minor cross-neutralizing antibody

**Fig. 7 Immunogenicity analysis of chVLPs in mice. a** Comparable neutralizing antibody titers for anti-HPV16 and **b** anti-HPV52 neutralizing antibody titers elicited by HPV16L1-C175A–HPV52L1-C428A chVLPs (orange column) and HPV16/52 VLPs as a mix (aqua column), administered in serial dosages in mice. **c–l** Mouse serum following immunization with various chVLPs and the corresponding WT VLPs mixture were evaluated using a pseudovirus-based neutralization assay (PBNA) with 10 HPV types. These chVLPs elicited 1–3 heterologous type neutralizing antibodies (violet) that were observably higher than that elicited by the WT VLPs (dark green), while maintaining comparable homo-type neutralizing antibody elicitation (orange) with respect to VLP mixture (aqua). For **a–c** $n = 4$ mice in each group; for **d–l** $n = 5$ mice in each group. The neutralization titers were presented as mean ± SD. Statistical significance was assessed by two-way ANOVA: *$P < 0.05$; **$P < 0.01$; ***$P < 0.001$; ****$P < 0.0001$. Non-neutralization samples were assigned a value of half of the limit of detection for visualization (the dotted line). Source data and exact $p$ values are provided as a source data file.

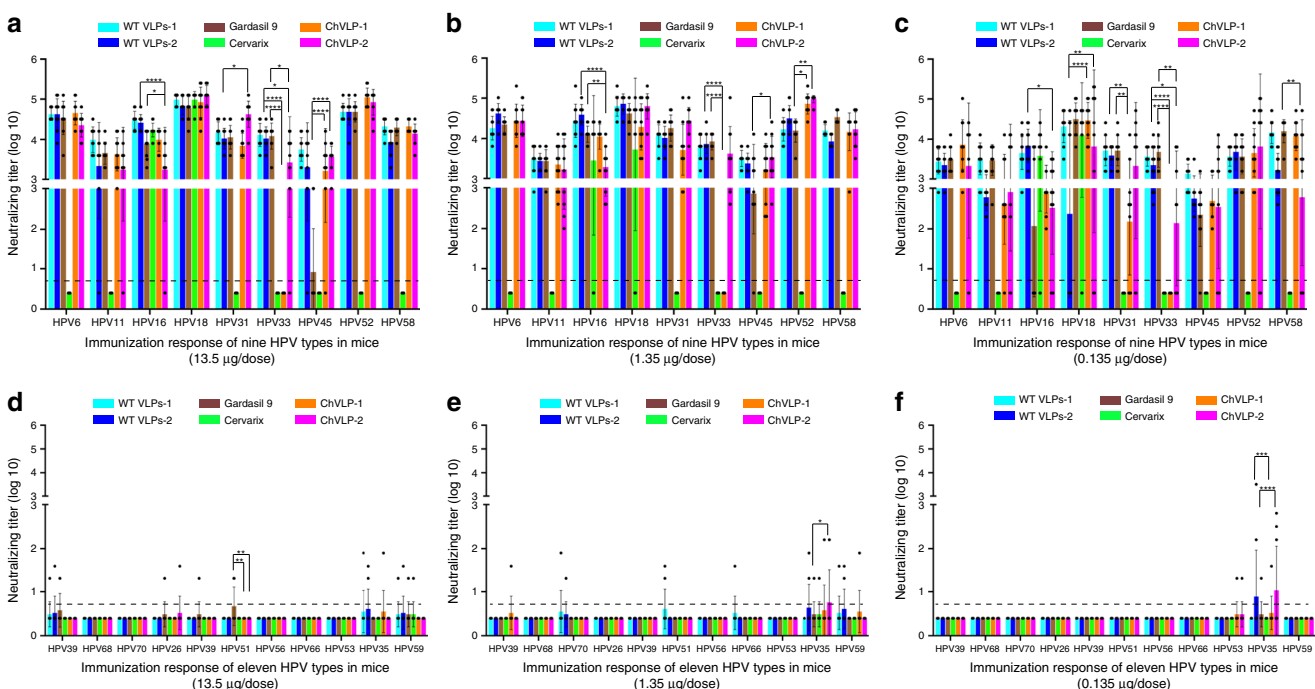

**Fig. 8 Immunogenicity analysis of a nine-valent chVLP vaccine delivered to mice.** Neutralization titers from mouse serum following immunization using WT VLPs-1 (cyan), WT VLPs-2 (blue), Gardasil 9 (brown), Cervarix (green), nona-type chVLP-1 (orange), and nona-type chVLP-2 (magenta). Titers were measured by PBNA covering the nine HPV types included in the vaccines (**a–c**) and the 11 non-vaccine HPV types (**d–f**). Eighteen groups of mice were immunized with three dosages (13.5 µg, 1.35 µg, and 0.135 µg) of the six different vaccines at 0, 2, and 4 weeks. Serum samples were collected 2 weeks after the third immunization. $N = 10$ mice in each group. The neutralization titers were presented as mean ± SD. Statistical significance was assessed by two-way ANOVA: *$P < 0.05$; **$P < 0.01$; ***$P < 0.001$; ****$P < 0.0001$. Non-neutralization samples were assigned a value of half of the limit of detection for visualization (the dotted line). Source data and exact $p$ values are provided as a source data file.

titers against heterologous HPV26, −35, −39, −53, and −59 in some of the mice (Fig. 8). Unexpectedly, one form of the nona-type chVLP (chVLP-1) did not induce detectable neutralizing antibody titers against HPV33 (Fig. 8a–c), although HPV33 L1 protein could be detected in the chVLP-1 particles (Supplementary Fig. 14b, d), indicating a different assembly modality for certain genotype of L1 between C175A and C428A mutants involved in the chVLP assembly. Taken together, chVLPs retain the same high level of immunogenicity as the prototypical HPVs and present minor cross-neutralization toward other heterologous types of HPV.

**Discussion**

HPV L1 proteins assemble into pentameric capsomeres that combine to form a $T = 7$ icosahedral lattice of 72 pentamers, without the involvement of hexamers, a geometry that is distinct from that of canonical icosahedron assembly[42]. It has been surmised that these pentamers serve as local blocks similar to hexamers in a so-called quasi-6-fold-equivalent manner during particle assembly in addition to presenting at the 5-fold symmetric apexes of the icosahedron[2,24]. Through structural analysis of the full atomic models of the HPV16 PsV (PDB no.

5KEQ) and HPV59 VLP (PDB no. 5JB1), we find that the interfacing regions at pentamer contacts (including between the 5-ordinated pentamers and 6-ordinated pentamers, or between two 6-ordinated pentamers) range from 77.7 Å² to 167.1 Å² (as calculated by the PISA server[43]), which suggests that these recognitions, while initiating self-assembly, have lower specificity as well as weak and mostly transient interactions, as assessed by the criteria for the formation of stable protein complexes in structural biology[6,40,44]. Thanks to the indispensable and conserved disulfide bonds between C175 and C428 (HPV16 numbering), these weak interactions in HPV capsids are stapled together to stabilize capsid assembly. Here, we verified the indispensability of the S–S linkage and the weaker interactions of the suspended bridges and their counterparts using saturated point mutations on both C175 and C428, and, intriguingly, no particles could be resolved in HPSEC or observed in TEM for any of the 38 mutants (Supplementary Figs. 15 and 16). When we combined the mutants in solution, however, we found that particle assembly was dictated by molar ratios of the mutants, and that more stable and homogeneous assembled particles (i.e., higher resolution cryo-EM structures) were associated with a higher proportion

of disulfide bonds during assembly, and, even more so, during maturation (for a maximum number of 720 bonds)[6].

Based on the well-defined mechanism of HPV L1 capsid assembly established in previous work[27–29], we propose that capsomere-hybrid particle assembly maintains the following attributes: (1) A single disulfide bond between the non-mutated C175 or C428 of alternate mutants still facilitates the linkage of all capsomeres and their arrangement into a $T = 7$ icosahedron (Supplementary Fig. 17), resulting in similar favorable morphology for chVLPs as that of the prototypic VLPs (Fig. 3e); (2) A 1:1 molar ratio of the involved C175A and C428A L1 proteins is favorable for hybrid-assembly (Fig. 2a), as this ratio maximizes the number of pairs of reciprocal C175 and C428 binding, and gives a maximum number of 360 bonds; and (3) The random pairing manner independent of the sequences of the various HPV genotypes (Fig. 1), which results from a lower specific recognition at the pentamer–pentamer interface and allows more genotypes of L1 pentamers to be incorporated into single particles. We confirm at least three genotypes can be incorporated into single particles (Supplementary Fig. 12), tested a total of nine types for co-assembly, and propose 72 types as a maximum of theoretical randomization.

The major neutralization sites that determine conformation predominantly localize to the five surface loops of HPV L1s, and are generally irrelevant to capsid assembly[45]; albeit, in rare reports, suspended bridge regions contribute to neutralization determinants, such as in the anti-HPV16 U4 epitope[40] and the anti-HPV59 CTA epitope[6]. In the antigenicity analysis of chVLPs, we found co-assembly disturbed the binding of several type-specific neutralizing antibodies to chVLPs to an extent ranging from 1- to 4-log as compared with the WT VLPs. Moreover, few mAbs, such as chVLP-specific mAb 10C3 and anti-HPV59 mAb 13A6, only reacted with HPV16/52 di-type chVLPs instead of their parental-type VLPs (Fig. 6c), which suggests that the hybrid assembly may slightly alter the original antigenicity that would stem from the prototypic pentamers or VLPs. Hybrid-assembly might also create some cross-type neutralization sites on the inter-pentameric interface, given that the interface is constituted by the two stretches of L1 sequences from the two parental types via the reciprocal linkage of the unmutated cysteine residues. Interestingly, although the antigenicity of various chVLPs was altered, potent antibody elicitation for the prototypical neutralization was still maintained and generated some cross-type protection. It is reasonable that the cross-neutralizing antibody response is limited for chVLPs, due to that the immunodominant epitopes of HPV L1 are mainly located on the five surface loops[46–48]. Furthermore, the suspended bridge region, which varies in the capsomere-hybrid co-assembly, may have a lower immunogenicity when considered in the context of the surface loops that constitute most of immunodominant epitopes. Of note, the cross-neutralizing antibody titer against heterologous types (Fig. 8d–f) induced by nona-type chVLPs seems like much lower than that of di- to penta-type chVLPs (Fig. 7c–l), possibly due to more obvious immune interference for cross-neutralization in more valent vaccine[49]. Therefore, the cross-type neutralization of chVLPs should be further tested, with the assistance of some stronger adjuvants, such as AS04[50], in non-human primates. Nonetheless, the nona-type chVLP has some competitive advantages over Gardasil 9 in the aspect of practical vaccine development, including a more stable pentamer stage in the purification process, a more controllable assembly without the requirement for a reductant (e.g., Dithiothreitol (DTT)) during assembly, and a one-time formulation. However, while we have used an established surrogate assay (PBNA) for assessing protection induced by the HPV nine-valent chVLP vaccines, protection should be further tested in an in vivo challenge model.

Structural vaccinology is an emerging field that aims to redesign immunogens with improved antigenicity and a degree of immunogenicity that facilitates broad-protective neutralization. Previously, we used rational design to create a tri-type HPV vaccine candidate by loop-swapping of the immunogenic surface loops[51]. Here, our chVLP design allows multiple pentamer types (9 types tested in the present study, 72 types in theory) to assemble into single VLPs, which can, in turn, dramatically decrease the number of VLPs required for the development of a pan-HPV vaccine. Furthermore, we surmise that a combination of these two strategies might facilitate the development of a pan-HPV vaccine that could theoretically incorporate 216 types of HPV immunodominant epitopes into a single particle (i.e., 72 pentamers, each bearing tri-type immunogenic loop regions). The number of types covered in the chHPV design approximates the number of HPV genotypes identified to date. Furthermore, in addition to bearing the merits of a traditional VLP for vaccine design, the chVLP offers a better capacity to harbor considerable epitopes or foreign peptides, such as various neoantigens for personal cancer therapeutic vaccines[22,52]; may provide an ideal transport cargo with controllable assembly; and may offer abundant surface decoration potential for drug delivery.

## Methods

**Constructs and strain construction.** HPV6 (GenBank: AAC80442.1), HPV11 (AAA46935.1), HPV16 (ANA05496), HPV18 (AAQ92369), HPV31 (P17388), HPV26 (NP_041787.1), HPV33 (AMY16565), HPV35 (P27232), HPV39 (P24838), HPV45 (P36741), HPV51 (ACV88631.1), HPV52 (AML80965), HPV53 (NP_041848), HPV56 (P36743), HPV58 (AFS33402), HPV59 (CAA54856), HPV66 (ABO76893), HPV68 (AGU90787), HPV69 (AHV83654.1), and HPV70 (P50793) L1 genes were used for L1 expression and pseudovirus preparation. N-terminally truncated HPV6, –11, –16, –18, –31, –33, –35, –45, –52, –58, and –59 L1 genes were cloned into pTO-T7 vector[30]. A series of site-directed mutations on HPV L1-C175 and -C428 were created using the Fast Mutagenesis Kit (Vazyme, Nanjing, China). These mutated HPV L1 genes were cloned into the pTO-T7 expression vector[53] and the E.coli ER2566 strain was used for protein expression. The HPV52 L1/L2 genes were synthesized by GLS (GL Biochem, Shanghai, China). All the C175A/C428A mutations in the L1 genes were generated using the Fast Mutagenesis Kit. The phylogenetics of HPV L1 proteins was analyzed by MEGA 10.1.7.

**Protein purification and particle assembly.** HPV WT and mutant L1 proteins were produced in ER2566 E. coli strain[6,30,33,37]. In brief, cells were grown in LB medium at 37 °C until reaching an $OD_{600}$ of 0.6. L1 protein expression was induced by the addition of isopropyl-β-D-thiogalactoside (IPTG, final concentration of 10 μM) with the cells further incubated at 25 °C for 8 h. Cells were harvested by centrifugation and resuspended with cell lysis solution (20 mM Tris, pH 7.4, 300 mM NaCl, and 5 mM EDTA). HPV L1 proteins were released from the cells and combined with 20 mM DTT, and then further purified by SP sepharose (GE Healthcare, America) and CHT-II resin (Bio-Rad, America). After purification, the proteins were analyzed using SDS-PAGE and stored at a final concentration of 1 mg/mL in a solution containing 20 mM phosphate buffer, pH 8.0 (PB8.0), 500 mM NaCl, and 20 mM DTT. Single (WT) or mixed L1 proteins were then changed to assembly buffer (10 mM phosphate buffer, pH 6.5, 500 mM NaCl) to allow for particle assembly in vitro at ambient temperature.

**Capsomere-hybrid VLPs.** After purification, the BCA protein assay kit was used to determine the concentration of HPVL1-C175A and HPVL1-C428A mutants. For di-type chVLP, HPVL1-C175A and HPVL1-C428A mutants were mixed in an equimolar ratio and assembled in a buffer containing 10 mM PB6.5 and 0.5 M NaCl. After 12 h, the samples were purified on a Superdex 200 10/300GL column using an AKTA Explorer 100 (GE Healthcare, America) to collect chVLPs, which were stored at 4 °C. For multi-type chVLPs, we followed the same equimolar principle for the content of L1-C175A and -C428A mutants. The chVLPs names and multi-type chVLPs formations are shown in Supplementary Table 2.

**Murine monoclonal antibodies.** HPV L1-specific mouse mAbs were produced using hybridoma technology[54,55]. BALB/c mice were primely immunized subcutaneously with HPV VLPs formulated with Freund's complete adjuvant (50 μg/dose) at week 0 and then two boost immunizations using Freund's incomplete adjuvant were implemented at weeks 2 and 4. The anti-HPV L1 VLP and chVLP mAbs were produced from mouse ascites fluid and were screened by HPV VLP-based ELISA and a pseudovirus-based neutralization assay (PBNA). Protein A affinity chromatography was used to purify anti-HPV mAbs IgGs. The purified

mAbs were diluted to 1.0 mg/mL and stored in phosphate-buffered saline (PBS) at −20 °C.

**SDS-PAGE and western blotting**. SDS-PAGE was performed using the Laemmli method with minor modifications[56]. Protein samples were mixed with equal volumes of loading buffer (100 mM Tris-HCl pH 6.8, 200 mM BME, 4% SDS, 0.2% bromophenol blue and 20% glycerol), heated at 80 °C for 10 min, and then loaded into the wells of a 10% separating gel.

For western blotting, separated HPV L1 proteins were transferred to nitrocellulose membranes and blocked with 5% skim milk. After blocking, membranes were incubated with HPV16- or HPV52-L1-specific mAbs (21A5 or 12D10; 1:1000 dilution) at room temperature for 1 h, and then washed with PBS (pH 7.4) containing 0.2% Tween-20. Membranes were then incubated with goat anti-mouse alkaline phosphatase-conjugated antibodies (Abcam; Cambridge, UK; 1:5000 dilution) followed by color development for 5 min using NBT/BCIP reagent (Pierce Biotechnology; Rockford, IL).

**Transmission electron microscopy**. Approximately 15 μL of negatively stained WT VLPs and chVLPs (100 μg/mL) were absorbed onto carbon-coated copper grids, blotted dry, and stained with freshly filtered 2% phosphotungstic acid (pH 6.4). Grids were examined using an FEI Tecnai T12 TEM at an accelerating voltage of 120 kV, and photographed at a nominal ×25,000 magnification.

**Analytical ultra-centrifugation**. Sedimentation velocity experiments of WT VLPs and chVLPs were carried out at 20 °C on a Beckman XL-A analytical ultra-centrifuge, equipped with absorbance optics and an An60-Ti rotor (Beckman Coulter; Fullerton, CA). Samples were diluted to 0.8 mg/mL (~1.0 OD$_{280nm}$) in 10 mM PB 6.5 with 0.5 M NaCl. The rotor speed was set to 7000 rpm for the highest resolution. The sedimentation coefficient was obtained using the c(s) method with Sedfit software[57], kindly provided by Dr. P. Schuck at the National Institutes of Health (Bethesda, MD).

**Size-exclusion chromatography (HPSEC)**. The homogeneity of the WT VLPs and chVLPs was determined by HPSEC (Agilent Technologies 1200 series; Santa Clara, CA) through a TSK Gel G 5000 pwxl 7.8 × 300 mm column (TOSOH, Tokyo, Japan) equilibrated in 10 mM PB, pH 6.5, containing 0.5 M NaCl. The column flow rate was maintained at 0.5 mL/min and proteins in the eluents were detected at 280 nm.

**Dynamic light scattering**. The hydrodynamic size distributions of WT and chVLPs were measured using a NanoBrook Omni DLS system. Samples (~50 μL) were loaded to overfill a 50-μL plastic sample cell with a 10-mm path length. Data were collected and analyzed with dynamics software. The hydrodynamic radius (Rh) of the particles was calculated through the Stokes-Einstein equation and the mean of three separate acquisitions were computed.

**Differential scanning calorimetry**. The thermostability of WT VLPs, chVLPs, and HPV pentamers were determined using a MicroCal VP-Capillary DSC (GE Healthcare, MicroCal Products Group, Northampton, MA)[58]. In brief, samples were diluted to 0.5 mg/mL in 10 mM PB, pH 6.5, containing 500 mM NaCl, and measured at a scanning rate of 1.5 °C/min, with the scan temperature ranging from 10 °C to 90 °C. Melting temperatures (Tm) were calculated using MicroCal Origin 7.0 (Origin-Lab Corp., Northampton, MA) software assuming a non-two-state unfolding model.

**Cryo-electron microscopy**. Purified HPV16 VLPs, HPV52 VLPs, and HPV16L1-C175A–HPV52L1-C428A chVLPs (~2.0 mg/mL) were vitrified on Quantifoil holey carbon grids in an FEI Vitrobot[6]. Images were recorded on an FEI Falcon II direct detector camera at a nominal ×93,000 magnification in an FEI TF30 FEG microscope at 300 kV, with underfocus settings estimated to be between 1.0 and 3.0 μm, and an electron dose of 25 e/Å$^2$. Data were automatically collected using FEI EPU. Drift and beam-induced motion correction were performed with MotionCor2[59]. Phase-shift estimation were carried out with Gctf[60]. The $T = 7$ particle images were manually boxed and extracted with the program Robem[61]. Relion 2.0, AUTO3-DEM, and EMAN2 programs were used for image processing and three-dimensional (3D) reconstructions[62,63]. All figures were generated by Chimera[64].

**Disulfide bond analysis**. The total or free sulfhydryl (SH) content of HPV16L1-C175A–HPV52L1-C428A chVLPs and WT VLPs was determined according to the method of Yongsawatdigul and Park[31,32]. For the total SH content, samples were serially diluted to a final L1 concentration of 0.2, 0.4, 0.6, or 0.8 mg/mL with solubilizing buffer (0.086 M Tris-HCl, 0.09 M glycine, 0.04 M EDTA, 8 M urea, pH 8.0). For the free SH content, samples were diluted in the same manner but with urea-free solubilizing buffer (0.086 M Tris-HCl, 0.09 M glycine, 0.04 M EDTA, pH 8.0). Each diluted sample (5 mL) was then mixed with 50 μL Ellman's reagent (2 mM DTNB in 0.2 M Tris-HCl, pH 8.0), and the mixtures were incubated at 25 °C for 40 min. The SH content was calculated from the absorbance measured at

412 nm using a molar extinction coefficient of 13,600 M$^{-1}$ cm$^{-1}$. A simple linear regression model was applied to optimally fit the correlation between the concentration of VLPs and the absorbance values. The disulfide bond concentrations for chVLPs and WT VLPs were calculated using the equations Csh = 73.53 × (A/C) and Cdb = (Csht−Cshf)/2, where Csh is the concentration (μmol/g) of the SH content, A is the absorbance value at 412 nm, C is the concentration of the samples, Csht and Cshf are the concentration (μmol/g) of the total SH content and free SH content, respectively, and Cdb is the concentration (μmol/g) of the disulfide bond content of the samples. Each sample was analyzed in triplicate.

**Enzyme-linked immunosorbent assay**. The wells of 96-well microplates were coated with WT VLPs, hybrid HPV L1 VLPs, HPV pentamers (100 ng per well) or monoclonal antibodies (200 ng per well). The wells were blocked with blocking solution and incubated with 100 μL of 2-fold serially diluted anti-HPV WT L1, hybrid L1 mAbs or antigen (pentamer, VLPs) with the start concentration of 1 μg/mL. The wells were washed and then incubated with HRP-conjugated goat anti-mouse IgG antibody (Abcam; Cambridge, UK; 1:5000 dilution) or the secondary mAb diluted 1:5000 in HS-PBS. Following this, the wells were incubated with 50 μL 3, 3′, 5, 5′-tetramethylbenzidine for 10 min at 37 °C. The reactions were quenched with 50 μL of 2 mol/L H$_2$SO$_4$, and the absorbance at 450 nm (620 nm reference) was recorded using an automated ELISA reader (TECAN, Männedorf, Switzerland). The cut-off value for positive titers was set to an absorbance of 0.1. The reactivities of various VLPs were represented by EC$_{50}$ values, which reflects the antibody concentration required to achieve 50% of the maximal signal.

**Preparation of HPV pseudoviruses**. HPV 6, −11, −16, −18, −26, −31, −33, −35, −39, −45, −51, −52, −53, −56, −58, −59, −66, −68, −69, and −70 pseudoviruses (PsVs) were produced in 293FT cells[65–67]. The HPV16 L1 and L2 expression vector, and the pN31-EGFP used in the experiment were kindly provided by Dr. J. T. Schiller[34]. 293FT cells were harvested at 72 h after transfection, lysed in a Dulbecco's PBS-Mg solution cell lysis buffer comprising 0.5% Brij58 (Sigma-Aldrich), 0.2% Benzonase (Merck), 0.2% PlasmidSafe ATP-Dependent DNase (EPICENTRE Biotechnologies), and incubated at 37 °C for 24 h. Afterwards, 5 M NaCl solution was added to the samples to extract the cell lysate. The tissue culture infective dose (TCID$_{50}$) of the supernatant was then measured to determine the titers of the PsVs, calculated according to the classical Reed-Muench method[68]. For chPsVs, the mutant pAAV-HPV16L1-C175A and pAAV-HPV52L1-C428A plasmids were mixed in equal proportions for transfection into 293FT cells. These PsVs were purified using a Capto Core 700 (GE Healthcare, America). After purification, the PsVs were analyzed by TEM.

**Pseudovirus-based neutralization assay**. A pseudovirus-binding neutralizing assay was used to determine anti-HPV L1 neutralizing antibodies within the samples[66,67]. Briefly, 293FT cells were incubated at 37 °C in the wells of a 96-well plate at a density of $1.5 × 10^4$ cells per well for 6 h. Sera were diluted according to a 2-fold dilution, and PsVs were diluted to $3 × 10^5$ TCID$_{50}$/μL. Equal volumes (60 μL) of the PsV diluent and the serially diluted sera were mixed and incubated at 4 °C for 1 h. The negative control was prepared by mixing equal volumes (60 μL) of the PsV diluent and culture medium. Then, 100 μL of these mixtures were added to designated wells and incubated at 37 °C for 72 h. The endpoint titers were calculated as the log$_{10}$ of the highest sera or antibody dilution with a percent infection inhibition higher than 50%. The resulting datasets were statistically analyzed using software Prism 7.0.

**Ethics statement**. In our study, all animal experimental protocols were reviewed and approved by the Animal Care and Use Committee of Xiamen University. The manipulation and vaccination of animals strictly adhered to and complied with the guidelines provided by XMULAC. All efforts were made to minimize suffering during vaccination, blood collection, and surgery. Finally, experimental animals were injected with nembutal sodium for euthanasia.

**Vaccines**. To formulate the multi-valent HPV VLP vaccine, each VLP type was first individually absorbed to an aluminum hydroxide adjuvant, and then mixed together, resulting in a final amount of 0.42 mg aluminum hydroxide suspended in 1.0 mL solutions. For the multi-type chVLPs, each chVLP was formulated with aluminum hydroxide adjuvant with an equivalent amount of L1 and aluminum hydroxide as that in the multi-type HPV VLPs in 1.0 mL solutions. The commercial HPV vaccines, Gardasil 9 (Lot no. N023354) and Cevarix (Lot no. S007151) were purchased (Hong Kong) and diluted with aluminum hydroxide adjuvant solution, according to intended antigen amount, to serve as controls. The 9-valent WT VLP vaccine controls have same VLP formulation as Gardasil 9 but in two final volumes, 0.5 mL (WT VLPs-1) and 1.0 mL (WT VLPs-2).

**Animals, immunizations, and serological analysis**. BALB/c female mice were maintained in a temperature-controlled facility with 12 h light-dark cycle. To initially assess the immunogenicity of HPV16L1-C175A–HPV52L1-C428A chVLPs, BALB/c mice ($n = 4$) were immunized intraperitoneally three times at an interval of 2 weeks (weeks 0, 2, and 4) with chVLPs or WT VLPs diluted in

aluminum adjuvant (5.0 µg per dose). Serum samples were collected at week 8, and the neutralizing titers were detected using the PBNA, as described above.

For dose-dependent response immunogenicity evaluations, BALB/c mice ($n = 4$ per dose) were immunized intraperitoneally three times at an interval of 2 weeks (week 0, 2, and 4) with 5, 1, 0.2, 0.04, 0.008, or 0.002 µg dosages of HPV16L1-C175A–HPV52L1-C428A chVLPs or WT VLPs along with aluminum hydroxide adjuvant. Serum samples were collected at week 8 to determine the neutralizing titers by the PBNA.

To assess the immunogenicity of multi-type chVLPs, BALB/c mice ($n = 5$) were immunized intraperitoneally three times at an interval of 2 weeks (weeks 0, 2, and 4) with tri-type (2.5 µg per dose), tetra-type (3.5 µg per dose), penta-type (4.0 µg per dose), hexa-type (5.0 µg per dose), or hepta-type (5.0 µg per dose) chVLPs and WT VLPs diluted in aluminum adjuvant. The immunizing doses of the multi-type chVLPs and the content of each HPV L1 type are shown in Supplementary Tables 2 and 3. The compositions and concentrations of the multi-type WT VLPs are the same as that for the multi-type chVLPs. Serum samples were taken at week 8 after the first immunization to determine the neutralizing titers by PBNA. For the immunogenicity assay of the nona-type chVLPs, BALB/c mice ($n = 10$) were also immunized intraperitoneally three times at an interval of 2 weeks (week 0, 2 and 4) and sera were harvested at week 6 after the first immunization. Three dosages (13.5 µg, 1.35 µg and 0.135 µg) were used in the nona-type chVLP groups and control ones (WT VLPs-1, WT VLPs-2, Gardasil 9, Cervarix). The detailed formulation is shown in Supplementary Table 4.

**Reporting summary**. Further information on research design is available in the Nature Research Reporting Summary linked to this article.

## Data availability

All data supporting the findings of this study are available from the corresponding author on reasonable request. The EM density map for HPV16L1-C175A–HPV52L1-C428 chVLP is deposited in the Electron Microscopy Data Bank (accession code EMD-0878, https://www.ebi.ac.uk/pdbe/entry/emdb/EMD-0878). The source data underlying Figs. 6, 7, and 8, and Supplementary Figs. 3c, 10, and 11 are provided as a Source Data file. Source data are provided with this paper.

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

## Acknowledgements

This work was supported by grants from the National Natural Science Foundation (grant no. U1705283, 31670935, 81701637, 81971932, 81991491, and 31730029), National Key Projects in Science and Technology (grant nos. 2018ZX09738008).

## Author contributions

S.L., Y.G., and N.X. designed the study. D.W., X.L., M.W., C.Q., S.S., J.C., Z.W., Q.X., Y.Y., M.H., X.C., S.H., T.L., Z.K., and Q.Zheng performed experiments. D.W., X.L., M.W., H.Y., Y.W., Q.Zhao, J.Z., Y.G., S.L., and N.X. analyzed data. D.W., S.L., and Y.G. wrote the manuscript. D.W., X.L., M.W., H.Y., Y.W., Q.Zhao, J.Z., S.L., Y.G., and N.X. participated in discussion and interpretation of the results. All authors contributed to experimental design.

## Competing interests

The authors declare no competing interests.

## Additional information

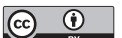

