## [Peer Review File · Nature Communications]

Reviewers' Comments:

Reviewer #1:

Remarks to the Author:

The major claim is that chimeric HPV VLPs can induce neutralizing responses to multiple HPV types. The approach to produce the chimeric HPV VLPs is quite novel and different from the previous designs to produce HPV VLPs that induce neutralizing antibodies to multiple HPV types using HPV peptides from L2.

The chimeric HPV VLPs were well characterized in vitro and the in vivo study demonstrated that they induce neutralizing antibodies to HPV types included in the chimeric VLPs. These titers were not significantly different from native VLPs. However, the cross neutralizing antibody responses were limited to a few types so that a pan vaccine design with chimeric HPV VLPs would need to include the most prevalent HPV types.

The quality of the writing including the precision of the description about HPV mechanism of infection and inaccurate terminology (eg hetero-type neutralizing antibodies) should be addressed. The value of the scientific findings are difficult to appreciate due to especially the way they are described in the abstract.

Additional comments:

- Line 45 in introduction incorrectly states that vaccines only induced type specific protection.
- In vitro antigenicity work demonstrated the loss of some type specific mAb binding, but is not clear how the authors claim that "some new type-cross neutralization" (line 325).
- Despite the extensive in vitro evaluation there is no data on the stability of the chimeric VLPs, given 1/2 the disulfide bonds, vs non-chimeric VLPs.
- Lack of detail about the formulation of the vaccine with aluminum, control for adsorption etc.
- No information on the statistical analysis: eg Figure 7 asterisk indicate significance ?
- Not clear of the added value of Figure 6C: generation of mAb to ChVLPs ?
- Methodology for Figure S3 not very clear ?

Reviewer #2:

Remarks to the Author:

This is an interesting study using a novel strategy generating capsomere-hybrid HPV L1 VLP to develop HPV vaccine candidates that confer broad-spectrum protection.

The authors rationally design single C175A or C428A L1 mutants that assemble into pentamers, but are deficient for VLP assembly. Mixing C175A and C428A L1 mutants of different HPV types in equimolar ratios hybrid VLP are generated that mimic wild-type L1 VLP morphologically and immunologically. Up to seven types are shown to co-assemble, and the authors propose a maximum of 72 types could co-assemble.

The hybrid VLP were extensively characterized by biophysical and biochemical means for particulate, structural characteristics, thermal stability.

Although the HPV16-containing hybrid VLP retained the immunodominant neutralization epitope recognized by mAb V5, some hybrid VLP lost high-affinity binding sites to mAb.

In contrast to a mixture of VLP that induced antisera with neutralization of the included (homologous) VLP as expected, the hybrid VLP containing the same mixture of capsomers of HPV types were also highly immunogenic and in addition induced cross-neutralization to heterologous types (e.g. the hepta-type hybrid VLP induced neutralization against 10 HPV types).

The authors claim to be able to develop an improved HPV vaccine with better cross-protection than licensed multivalent HPV vaccines. However, it remains unclear, if a multivalent combination of capsomers into hybrid VLP (as in this study) has a competitive advantage over a combination of wild-type L1 VLP (e.g. Gardasil-9). Also, Cervarix and Gardasil vaccines induce some cross-neutralization and cross-protection.

Major point:

This study would be much stronger if the experimental chVLP vaccine efficacy were tested (using Gardasil-9 in comparison) also in vivo in the mouse vaginal model (or rabbit cutaneous model) against challenge with (homologous and heterologous) HPV pseudovirion types for which neutralization is claimed.

Minor points:

Line 44 they induce mostly type-restricted neutralizing antibodies

Line 95 transmission electron microscopy (TEM)

Line 102 explain HPSEC

Line 127 explain AUC-SV

Line 233: mAb HPV16.V5 is directed to conformational epitope (also Fig. 6)

Line 237:immunization of mice with HPV16L1-C175A-.....

Line 238 and line 325: 2F4 and 3A6 also target HPV16 VLP, and there is also little specificity for 10C3

Line 267: We then measured immune sera against 10 genotypes....

Line 328: ,Hybrid-assembly also alters the type-cross neutralization sites of the inter-pentamer interface. ...'. Please explain on which data this statement is based.

Line 332: ,broad type-cross protection' should be attenuated to ,some cross-type neutralization'.

Line 376: explain PB8.0

Line 377, 383: explain PB6.5

Line 381 ...bivalent chVLP

Line 388: Table S2

Line 409: Transmission electron microscopy (TEM)

Response to Reviewer Comments on the manuscript [NCOMMS-19-13881-T]:

We thank the two reviewers for recognizing the merit of our work and for their suggestions to improve our manuscript. We have fully addressed the comments with appropriate additional experiments and analyses. To facilitate the navigation of this document, we have copied the reviewers' comments verbatim in **blue** and typed our responses in **black**.

Summary of the revisions made, with new experimental data:

(1) A nine-valent chVLP with the same composition as Gardasil-9 was generated and subjected to an immunogenicity assay in mice, as compared to the immune response of WT VLP mixture, Gardasil 9, Cervarix HPV vaccines. The neutralizing antibody titers against the nine types of homologous HPV (HPV 6, -11, -16, -18, -31, -33, -45, -52, -58) are comparable with that of Gardasil-9, with minor type-cross neutralization against some of other 11 heterologous HPV types.

(2) The stability of a bivalent chVLP was characterized by a multi-faceted approach over a long-term incubation of 10 weeks at 4°C, 25°C and 37°C. We show that chVLPs and WT VLPs have comparable stability.

Reviewer #1

Comments to the Author

Reviewer: The major claim is that chimeric HPV VLPs can induce neutralizing responses to multiple HPV types. The approach to produce the chimeric HPV VLPs is quite novel and different from the previous designs to produce HPV VLPs that induce neutralizing antibodies to multiple HPV types using HPV peptides from L2.

The chimeric HPV VLPs were well characterized in vitro and the in vivo study demonstrated that they induce neutralizing antibodies to HPV types included in the chimeric VLPs. These titers were not significantly different from native VLPs. However, the cross neutralizing antibody responses were limited to a few types so that a pan vaccine design with chimeric HPV VLPs would need to include the most prevalent HPV types.

The quality of the writing including the precision of the description about HPV mechanism of infection and inaccurate terminology (eg hetero-type neutralizing antibodies) should be addressed. The value of the scientific findings are difficult to appreciate due to especially the way they are described in the abstract.

Response: We thank the reviewer for the encouraging comments on the novelty of our approach. In relation to the neutralization titers elicited by the chVLPs—as also suggested by the Reviewer #2—we include the following text explanation and new data:

“Finally, we generated a nine-valent chVLP according to the composition and dosage of L1s in the Gardasil-9 formulation, and measured the neutralizing antibody titers with 20 types of HPV PsVs—nine homologous types (HPV 6, -11, -16, -18, -31, -33, -45, -52, -58) and 11 heterologous ones (HPV 26, -35, -39, -53, -51, -56, -59, -66, -68, -69, -70). Consistent with other multiple-type chVLPs (Figure 5 and 7), the nine-valent chVLP showed good physiochemical and particle nature (Figure S14) and elicited a high neutralizing antibody response against the nine homologous types of HPV similar to that of the WT VLP mixture and Gardasil-9, moreover, additionally induced some minor cross-neutralizing antibody titers against heterologous HPV26, -35, -39, -53 and -59 in some of the mice (Figure 8).” (Page 14, line 306-315).

The limited cross-neutralizing antibody response in terms of both the antibody titer and the type-breadth of the chVLP vaccine may indicate the requirement for a formulation with a stronger adjuvant for higher neutralizing antibody elicitation. In this regard, we toned down the significance of the cross neutralization of the chVLP throughout the manuscript, and rephrased the title from “Capsomere-hybrid human papillomavirus virus-like particles elicit hetero-type neutralizing antibodies” to “Rational design of a multiple-valent human papillomavirus vaccine by capsomere-hybrid co-assembly of virus-like particles”.

Furthermore, we propose a possible strategy with regard to the pan-HPV vaccine design, as described in the Discussion, which is based on this study and our findings in previous work where we created a triple-type HPV vaccine candidate by loop-swapping of the immunogenic surface loops (Li Z, *et al. Nature Commun* 2018;9;5360): “Furthermore, we surmise that a combination of these two strategies might facilitate the development of a pan-HPV vaccine that could theoretically incorporate 216 types of HPV immunodominant epitopes into a single particle (i.e., 72 pentamers, each bearing triple-type immunogenic loop regions). The number of types covered in the chHPV design approximates the number of HPV genotypes identified to date.” (Page 18, line 392-397)

We believe that the chVLP rationale could pave the way for a new mode of vaccine design targeting multiple pathogenic variants or hyper-varied cancer cells. Please also refer to our response to Reviewer #2 general comment.

As pointed out, “hetero-type” is incorrect terminology and this has been rephrased as “heterologous type” throughout the manuscript. The abstract has been rewritten to reflect the scientific findings in our study:

“Abstract

The capsid of human papilloma virus (HPV) spontaneously arranges into a T=7 icosahedral particle with 72 L1 pentameric capsomeres associating via disulfide bonds between Cys175 and Cys428. Here, we designed a capsomere-hybrid virus-like particle (chVLP) to accommodate multiple types of L1 pentamers by the reciprocal assembly of single C175A and C428A L1 mutants, either of which alone encumbers L1 pentamer particle self-assembly. We show that co-assembly between any pair of C175A and C428A mutants across at least nine homologous HPV genotypes occurs at a preferred equal molar stoichiometry, irrespective of the type or number of L1 sequences. A nine-valent chVLP vaccine—formed through the structural clustering of homologous HPV epitopes—confers neutralization titers that are comparable with that of Gardasil-9 and elicits minor cross-neutralizing antibodies against some heterologous HPV types. These findings may pave the way for a new vaccine design that targets multiple pathogenic variants or hyper-varied cancer cells.” (Page 2, line 21-34).

Additional comments

Comment 1: Line 45 in introduction incorrectly states that vaccines only induced type specific protection.

Response: We apologize for this incorrect description. We have rephrased the sentence as:

“...they induce mostly type-restricted neutralizing antibodies and have limited cross-protection against a few of the non-vaccine types.” (Page 4, line 65-67). Please also refer to our responses to Reviewer #2, Comment 1.

Comment 2: In vitro antigenicity work demonstrated the loss of some type specific mAb binding, but is not clear how the authors claim that “some new type-cross neutralization” (line 325).

Response: As suggested, we have revised the text as follows:

“Moreover, few mAbs, such as chVLP-specific mAb 10C3 and anti-HPV59 mAb 13A6, only reacted with HPV16/52 bivalent chVLPs instead of their parental-type VLPs (Figure 6C), which suggests ...”(Page 17, line 365-367)

Comment 3: Despite the extensive in vitro evaluation there is no data on the stability of the chimeric VLPs, given ½ the disulfide bonds, vs non-chimeric VLPs.

Response: Indeed, stability is a key point of concern for translating chimeric VLPs into vaccines. We examined multiple attributes of HPV16L1-C175A–

HPV52L1-C428A chVLPs during storage at 4°C, 25°C, and 37°C for up to 10 weeks, specifically assessing sequence integrity with SDS-PAGE and WB, thermal stability with DSC, and size distribution with HPSEC and AUC. We further confirmed morphology with TEM and the hydrodynamic radius with DLS. The WT HPV16 and HPV52 VLPs were stored and examined in parallel.

“Despite of ½ disulfide bonds formed, the chVLPs exhibited comparable stability with that of WT VLPs during storage at 4°C, 25°C, and 37°C for up to 10 weeks, as assessed via protein integrity with SDS-PAGE and western blotting (Figure S4), thermal stability with DSC (Figure S5), sedimentation coefficient values with AUC (Figure S6), particle component retention time with HPSEC (Figure S7), hydrodynamic radius with DLS (Figure S8) and VLP morphology under TEM (Figure S9).” (Page 8, line 172-177)

Comment 4: Lack of detail about the formulation of the vaccine with aluminum, control for adsorption etc.

Response: We now provide detailed information in the Materials & Methods section regarding the formulation:

“To formulate the multiple-type HPV VLP vaccine, each VLP type was first individually absorbed to an aluminum hydroxide adjuvant, and then mixed together, resulting in a final amount of 0.42 mg aluminum hydroxide suspended in 0.5 or 1.0 mL solutions. For the multi-valent chVLPs, each chVLP was formulated with aluminum hydroxide adjuvant with an equivalent amount of L1 and aluminum hydroxide as that in the multiple-type HPV VLPs in 1.0 mL solutions. The commercial HPV vaccines, Gardasil-9 (Lot no. N023354) and Cevarix (Lot no. S007151) were purchased (Hong Kong) and diluted with aluminum hydroxide adjuvant solution, according to intended antigen amount, to serve as controls.” (Page 26, line 585-593)

Comment 5: No information on the statistical analysis: eg Figure 7 asterisk indicate significance?

Response: We have now included the statistical analyses, as follows: “The repeated neutralization titers were averaged to generate mean values and corresponding standard deviations (SDs). Non-neutralization samples were assigned a value of half of the limit of detection for visualization. Statistical significance was assessed by one-way ANOVA: *P < 0.05; **P < 0.01; ***P < 0.001; ****P < 0.0001.” (Page 38, line 964-967)

Comment 6: Not clear of the added value of Figure 6C: generation of mAb to ChVLPs ?

Response: To explore any potential variation in antigenicity upon chVLP assembly, we generated mAbs from mice immunized with chVLPs, and that found most neutralizing mAbs were type-specific, similar to their parental types. However, a few mAbs showed binding activities that varied between chVLPs and WT VLPs; for example, mAb 10C3 recognized only chVLPs. These findings suggest some variation in antigenicity in response to the reciprocal assembly of chVLPs. The sentences were rephrased as “Moreover, few mAbs, such as chVLP-specific mAb 10C3 and anti-HPV59 mAb 13A6, only reacted with HPV16/52 bivalent chVLPs instead of their parental-type VLPs (Figure 6C), which suggests that the hybrid assembly may slightly alter the original antigenicity that would stem from the prototypic pentamers or VLPs. Hybrid-assembly might also create some type-cross neutralization sites on the inter-pentameric interface, given that the interface is constituted by the two stretches of L1 sequences from the two parental types via the reciprocal linkage of the unmutated cysteine residues. Interestingly, although the antigenicity of various chVLPs was altered, potent antibody elicitation for the prototypical neutralization was still maintained and generated some type-cross protection.” (Page 17, line 365-374)

Comment 7: Methodology for Figure S3 not very clear?

Response: A detailed protocol for disulfide bond analysis has now been added to the Materials & Methods section: “The total or free sulfhydryl (SH) content of HPV16L1-C175A–HPV52L1-C428A chVLPs and WT VLPs was determined according to the method of Yongsawatdigul and Park. For the total SH content, samples were serially diluted to a final L1 concentration of 0.2, 0.4, 0.6, or 0.8 mg/mL with solubilizing buffer (0.086 M Tris-HCl, 0.09 M glycine, 0.04 M EDTA, 8 M urea, pH 8.0). For the free SH content, samples were diluted in the same manner but with urea-free solubilizing buffer (0.086 M Tris-HCl, 0.09 M glycine, 0.04 M EDTA, pH 8.0). Each diluted sample (5 mL) was then mixed with 50 μ L Ellman’s reagent (2 mM DTNB in 0.2 M Tris-HCl, pH 8.0), and the mixtures were incubated at 25°C for 40 min. The SH content was calculated from the absorbance measured at 412 nm using a molar extinction coefficient of 13,600 $M^{-1} \cdot cm^{-1}$. A simple linear regression model was applied to optimally fit the correlation between the concentration of VLPs and the absorbance values. The disulfide bond concentrations for chVLPs and WT VLPs were calculated using the equations $C_{sh}=73.53*(A/C)$ and $C_{db}=(C_{sht}-C_{shf})/2$, where C_{sh} is the concentration (μ mol/g) of the SH content, A is the absorbance value at 412 nm, C is the concentration of the samples, C_{sht} and C_{shf} are the concentration (μ mol/g) of the total SH content and free SH content, respectively, and C_{db} is the concentration (μ mol/g) of the disulfide bond content of the samples. Each sample was analyzed in triplicate.” (Page 23 , line 519-536)

Reviewer: 2

Comments to the Author

Reviewer: This is an interesting study using a novel strategy generating capsomere-hybrid HPV L1 VLP to develop HPV vaccine candidates that confer broad-spectrum protection.

The authors rationally design single C175A or C428A L1 mutants that assemble into pentamers, but are deficient for VLP assembly. Mixing C175A and C428A L1 mutants of different HPV types in equimolar ratios hybrid VLP are generated that mimic wild-type L1 VLP morphologically and immunologically. Up to seven types are shown to co-assemble, and the authors propose a maximum of 72 types could co-assemble.

The hybrid VLP were extensively characterized by biophysical and biochemical means for particulate, structural characteristics, thermal stability. Although the HPV16-containing hybrid VLP retained the immunodominant neutralization epitope recognized by mAb V5, some hybrid VLP lost high-affinity binding sites to mAb.

In contrast to a mixture of VLP that induced antisera with neutralization of the included (homologous) VLP as expected, the hybrid VLP containing the same mixture of capsomers of HPV types were also highly immunogenic and in addition induced cross-neutralization to heterologous types (e.g. the heptatype hybrid VLP induced neutralization against 10 HPV types).

The authors claim to be able to develop an improved HPV vaccine with better cross-protection than licensed multivalent HPV vaccines. However, it remains unclear, if a multivalent combination of capsomers into hybrid VLP (as in this study) has a competitive advantage over a combination of wild-type L1 VLP (e.g. Gardasil-9). Also, Cervarix and Gardasil vaccines induce some cross-neutralization and cross-protection.

Response: We thank the reviewer for the encouraging summary of our chVLP co-assembly strategy. As suggested, we now include a comparison of our chVLP with the commercial formulations that show equal neutralization as well as minor cross-neutralization:

“Finally, we generated a nine-valent chVLP according to the composition and dosage of L1s in the Gardasil-9 formulation, and measured the neutralizing antibody titers with 20 types of HPV PsVs—nine homologous types (HPV 6, -11, -16, -18, -31, -33, -45, -52, -58) and 11 heterologous ones (HPV 26, -35, -39, -53, -51, -56, -59, -66, -68, -69, -70). Consistent with other multiple-type chVLPs (Figure 5 and 7), the nine-valent chVLP showed good physiochemical and particle nature (Figure S14) and elicited a high neutralizing antibody response against the nine homologous types of HPV similar to that of the WT

VLP mixture and Gardasil-9, moreover, additionally induced some minor cross-neutralizing antibody titers against heterologous HPV 26, -35, -39, -53 and -59 in some of the mice (Figure 8).” (Page 14, line 306-315).

“It is reasonable that the cross-neutralizing antibody response is limited for chVLPs, due to that the immunodominant epitopes of HPV L1 are mainly located on the five surface loops. Furthermore, the “suspended bridge” region, which varies in the capsomere-hybrid co-assembly, may have a lower immunogenicity when considered in the context of the surface loops that constitute most of immunodominant epitopes. Therefore, the type-cross neutralization of chVLPs should be further tested, with the assistance of some stronger adjuvants, such as AS04, in non-human primates.” (Page 17, line 374-381). Please also refer to our comments in response to Reviewer #1 general comment. With this in mind, we toned down the significance of the cross neutralization of chVLPs throughout the manuscript, and rephrased the title as “Rational design of a multiple-valent human papillomavirus vaccine by capsomere-hybrid co-assembly of virus-like particles”.

In regards to the final comment made by the reviewer, we have removed our previous claim that, “This method provides a new approach that may aid in the development of an improved HPV vaccine with better cross-protection against the various HPV genotypes.” Instead, we include the following text in the Discussion: “Nonetheless, the nine-valent chVLP has additional competitive advantages over Gardasil 9 in terms of practical vaccine development, including a more stable pentamer stage in the purification process, a more controllable assembly without the requirement for a reductant (e.g., DTT) during assembly, and a “one-time” formulation.” (Page 17, line 381-385).

Furthermore, we propose a possible strategy with regard to the pan-HPV vaccine design based on the findings of this study and of our previous study (Li Z, *et al. Nature Commun* 2018;9:5360), as follows: “Furthermore, we surmise that a combination of these two strategies might facilitate the development of a pan-HPV vaccine that could theoretically incorporate 216 types of HPV immunodominant epitopes into a single particle (i.e., 72 pentamers, each bearing triple-type immunogenic loop regions). The number of types covered in the chHPV design approximates the number of HPV genotypes identified to date.” (Page 18, line 392-397).

We believe that the chVLP rationale could pave the way for a new mode of vaccine design, targeting multiple pathogenic variants or hyper-varied cancer cells.

Major point : This study would be much stronger if the experimental chVLP vaccine efficacy were tested (using Gardasil-9 in comparison) also in vivo in the mouse vaginal model (or rabbit cutaneous model) against challenge with

(homologous and heterologous) HPV pseudovirion types for which neutralization is claimed.

Response: We acknowledge that a mouse vaginal model using PsV challenge in vivo is a sensitive measure to test for the production of protective antibodies against HPV infection (*Longet et al. J Virol. 2011;85.24:13253-13259; Day et al. Clin. Vaccine Immunol. 2012:CVI-00139*) and particularly beneficial for testing L2-based vaccine development (*Wu et al., PLoS one 2011;6:e27141; Jagu et al., J Virol 2013;87:6127–6136*). However, such a model cannot sufficiently meet the demand for the high-throughput detection of serum neutralizing antibody titers (*Jagu et al. J Virol 2013;87.11:6127-6136*). Therefore, unfortunately, such in vivo animal models were not used for this study, and are beyond the scope of this work. Instead, WHO guidelines indicate that a PsV-based neutralization assay (PBNA) in a cell model is the “gold standard” for assessing the protective potential of antibodies induced by HPV L1 vaccines (*Dessy et al. Hum Vaccines 2008:4.6:425-434; World Health Organization Expert Committee on Biological Standardization. Guidelines to assure the quality, safety and efficacy of recombinant human papillomavirus virus-like particle vaccines. WHO, Geneva 2007; World Health Organization. WHO meeting on the standardization of HPV assays and the role of WHO HPV LabNet in supporting vaccine introduction. WHO, Geneva 2008*). As such, we previously used the high-throughput and robust PBNA to develop our HPV16/18 bivalent vaccine (launched in China recently), HPV 6/11 bivalent vaccine, HPV 9-valent vaccine, and HPV 20-valent vaccine (*Gu et al. Vaccine 2017;35:4637-4645; Pan et al. Vaccine 2017;35:3222-3231; Wei et al. Emerg Microbes Infect. 2018;26;7:160; Li et al. Nature Commun. 2018;9:5360*), as well as the 20-type neutralization assay carried out in this study.

Specific Comments:

Comment 1: Line 44: ...they induce mostly type-restricted neutralizing antibodies

Response: As suggested, we have rephrased the sentence as: “...they induce mostly type-restricted neutralizing antibodies and have limited cross-protection against a few of the non-vaccine types.” (Page 4, line 65-67) Please also refer to the Reviewer #1 Comment 1.

Comment 2: Line 95: ...transmisssion electron microscopy (TEM)

Response: This has been amended.

Comment 3: Line 102: ... explain HPSEC

Response: This has been amended. (Page 6, line 126)

Comment 4: Line 127:... explain AUC-SV

Response: This has been amended. (Page 7, line 152)

Comment 5: Line 237:immunization of mice with HPV16L1-C175A-.....

Response: This has been amended. (Page 13, line 268)

Comment 6: Line 238 and line 325: 2F4 and 3A6 also target HPV16 VLP, and the is also little specificity for 10C3

Response: We apologize for this error. We have rephrased the sentence as follows: "... found that mAb 10C3 specifically targets chVLPs." (Page 13, line 269) "Moreover, few mAbs, such as chVLP-specific mAb 10C3 and anti-HPV59 mAb 13A6, only reacted with HPV16/52 bivalent chVLPs instead of their parental-type VLPs (Figure 6C), which suggests ..." (Page 17, line 365-367) Please also refer to the Reviewer#1, Comment 2.

Comment 7: Line 267: We then measured immune sera against 10 genotypes....

Response: This has been amended. (Page 14, line 299)

Comment 8: Line 328: Hybrid-assembly also alters the type-cross neutralization sites of the inter-pentamer interface ...'. Please explain on which data this statement is based.

Response: We acknowledge that this statement is not supported by available structural evidence, but is based on the finding that some type-cross neutralization sites appear, and that the major difference in the surfaces between the chVLPs and the parental-type VLPs is determined by the reciprocal linkage constituted by the two L1 sequences. As such, we toned down the statement as follows: "Hybrid-assembly might also create some type-cross neutralization sites on the inter-pentameric interface, given that the interface is constituted by the two stretches of L1 sequences from the two parental types via the reciprocal linkage of the unmutated cysteine residues." (Page 17, line 369-372)

Comment 9: Line 332: ,broad type-cross protection' should be attenuated to ,some cross-type neutralization'.

Response: This has been amended. (Page 17, line 374)

Comment 10: Line 376: explain PB8.0

Response: This has been amended. "...a solution containing 20 mM phosphate buffer, pH 8.0 (PB8.0) ..." (Page 20, line 442)

Comment 11: Line 377, 383: explain PB6.5

Response: This has been amended. "... assembly buffer (10 mM phosphate buffer, pH 6.5, 500 mM NaCl) to allow..." (Page 20, line 443-444)

Comment 12: Line 381 ...bivalent chVLP

Response: This has been amended. (Page 20, line 448)

Comment 13: Line 388: Table S2

Response: This has been amended. (Page 21, line 454)

Comment 14: Line 409: Transmission electron microscopy (TEM)

Response: This has been amended. (Page 22, line 475)

Reviewers' Comments:

Reviewer #1:

Remarks to the Author:

The authors have addressed all my comments and have added additional data to support the responses. I have added a few comments/questions to the manuscript (see attachment below) that need clarification.

The real strength and innovation of work lies with the possibility to combine a 9 valent vaccine in 1 VLP formulation, not the capacity to induce cross neutralization/protection.

Reviewer #2:

Remarks to the Author:

The reviewer agrees with the authors that the PsV-based neutralization assay (PBNA) is an established surrogate assay to for assessing protection induces by HPV VLP vaccines, that has been used in immunobridging studies for alternate dosing schedules, bridging to age 26 years or younger, and biosimilar vaccines, with post-licensure surveillance confirming effectiveness (response to major point of reviewer 2).

However, by developing a new technology generating capsomer-hybrid VLP as broad spectrum vaccine candidate, it appears crucial to use the more stringent protection against experimental challenge with HPV pseudovirions in an established mouse model (or alternatively quasivirion challenge in a rabbit model) as proof of principle for protection (vaccine efficacy) and non-inferiority to licensed Gardasil-9. Thus the authors should go the extra mile and show claimed broad-spectrum vaccine efficacy in the challenge model, which is now readily available in many labs worldwide.

Response to Reviewer Comments on the manuscript:

We thank the editor and reviewers for the last round of comments.

Reviewer #1

Comments to the Author

Reviewer: The authors have addressed all my comments and have added additional data to support the responses. I have added a few comments/questions to the manuscript (see attachment below) that need clarification.

The real strength and innovation of work lies with the possibility to combine a 9 valent vaccine in 1 VLP formulation, not the capacity to induce cross neutralization/protection.

Response: We thank the reviewer for the conclusive remark and the further minor comments/questions, which helps us finalize our manuscript. To facilitate the navigation of this document, we have copied the reviewer's comments verbatim in **blue** from the change-tracked word file and typed our responses in **black**.

Additional comments

Comment 1: Line 13, 'hyper-varied', not clear what this means here?

Response: "hyper-varied cancer cells" is rephrased as "cancer cells bearing diverse neoantigens" (Page 2, line 13-14)

Comment 2: Line 28, 'However, despite this lower immune pressure, the phylogenetics of HPV is complex, and there has been a gradual accumulation of more than 200 distinct genotypes, most of which exhibit type-specific neutralization', this needs clarification.

Response: The sentence has been rephrased as, "In terms of this lower immune pressure and type-restricted neutralization mostly evoked in human antiviral immunity, the phylogenetics of HPV is complex, and there has been a gradual accumulation of more than 200 distinct genotypes." (Page 3, line 33-36)

Comment 3: Line 182, not appropriate, more like same genotype.

Response: Yes, the L2 gene used in pseudovirus preparation belongs to the same genotype as L1 mutant. The sentence now reads, "...with the assistance of the L2 protein same genotype as L1, either HPV16L1-C175A or HPV52L1-C428A alone could assemble into particles..." (Page 10, line 255-256)

Comment 4: Line 248, figure difficult to understand. What is rEC50? Not clear what X indicates PA1, PD4 etc... ?

Response: To probe the antigenicity variation upon chVLP assembly, the rEC_{50} was defined as the reciprocal of the EC_{50} value in tests against the EC_{50} of WT VLPs (Page 12, line 362). Thus, a larger rEC_{50} value suggests a higher reactivity for chVLPs against mAbs in test with respect to WT VLPs. X axis represents various mAbs. The legend of Fig. 6 has been detailed as, "Varied antigenicity of chVLPs. (a) HPV16-specific mAbs, (b) HPV52-specific mAbs and (c) mAbs raised from HPV16L1-C175A–HPV52L1-C428A chVLPs were used to probe epitope variation. The reactivities of VLPs or pentamer mutants against various mAbs were quantified as EC_{50} values (bottom panels), and their corresponding rEC_{50} values (defined as $1/(EC_{50} [\text{chVLPs}, \text{pentamers or other VLPs}]/EC_{50} [\text{HPV16 or HPV52 VLPs}])$) were shown in the upper panels. A larger rEC_{50} value suggests a higher reactivity for chVLPs against mAbs in test with respect to WT VLPs. The EC_{50} value was obtained from the mean value of two repeated data. Source data are provided as a source data file." (Page 40, line 1384-1436)

Comment 5: Line 258, 3 different routs of immunization cited in material and method, any justification should be mentioned in M&M.

Response: Sorry for the mistake. We have double checked our experiment record, in fact, in the serological analysis in this work, all mice were immunized via intraperitoneal injection. We have corrected this information in M&M. (Page 29, line 920; Page 29, line 925; Page 29, line 933)

Comment 6: Line 291, one of the 9V did not induce HPV33 neutra titers should mention this.

Response: As suggested, the point was added, “Unexpectedly, one form of the nona-type chVLP (chVLP-1) did not induce detectable neutralizing antibody titers against HPV33 (Fig. 8a, b, c), although HPV33 L1 protein could be detected in the chVLP-1 particles (Supplementary Fig. 14b, d), indicating a different assembly modality for certain genotype of L1 between C175A and C428A mutants involved in the chVLP assembly.” (Page 15, line 478-482)

Comment 7: Line 294, the levels are very low near the cutoff of assay, looks like a few outliers. Why may the levels be so much lower than in figure 7 with 2-5 valent vaccine ? Maybe interference. Should explain in the discussion.

Response: It is a critical point especially for the immunization of a multi-valent vaccine (such as HPV 9-valent vaccine) that immune interference is manifested by lower antibody titer for the same genotype antigen with same dosage formulated in a fewer valent vaccine (such as HPV 16/18 bivalent vaccine). We agree with the review that the immune interference takes effect on the chVLP antigens as well in this study. We added the point in the Discussion section, “Of note, the cross-neutralizing antibody titer against heterologous types (Fig. 8d, e, f) induced by nona-type chVLPs seems like much lower than that of di- to penta-type chVLPs (Fig. 7c-l), possibly due to more obvious immune interference for cross-neutralization in more valent vaccine.” (Page 18, line 575-578)

Comment 8: Line 433, with aluminum?

Response: The sentence has been rephrased as, “BALB/c mice were primely immunized subcutaneously with HPV VLPs formulated with Freund’s complete adjuvant (50 µg/dose) at week 0 and then two boost immunizations using Freund’s incomplete adjuvant were implemented at week 2 and 4.” (Page 22, line 724-727)

Comment 9: Line 565, dose of human vaccine used in mice missing: Gardasil, Gardasil 9 and Cervarix?

Response: The information has been added, “For the immunogenicity assay of the nona-type chVLPs, BALB/c mice ($n = 10$) were also immunized intraperitoneally three times at an interval of 2 weeks (week 0, 2 and 4) and sera were harvested at week 6 after the first immunization. Three dosages (13.5 µg, 1.35 µg and 0.135 µg) were used in the nona-type chVLP groups and control ones (WT VLPs-1, WT VLPs-2, Gardasil 9, Cervarix). The detailed formulation is shown in Supplementary Table 4.” (Page 29, line 932-937)

Comment 10: Line 574, schedule?

Response: Sorry for the unclear description. We have rephrased this sentence as: "...immunized intraperitoneally three times at an interval of 2 weeks (week 0, 2 and 4) with 5, 1, 0.2, 0.04, 0.008, or 0.002 µg dosages..." (Page 29, line 920)

Reviewer #2

Comments to the Author

Reviewer: The reviewer agrees with the authors that the PsV-based neutralization assay (PBNA) is an established surrogate assay to for assessing protection induces by HPV VLP vaccines, that has been used in immunobridging studies for alternate dosing schedules, bridging to age 26 years or younger, and biosimilar vaccines, with post-licensure surveillance confirming effectiveness (response to major point of reviewer 2).

However, by developing a new technology generating capsomer-hybrid VLP as broad spectrum vaccine candidate, it appears crucial to use the more stringent protection against experimental challenge with HPV pseudovirions in an established mouse model (or alternatively quasivirion challenge in a rabbit model) as proof of principle for protection (vaccine efficacy) and non-inferiority to licensed Gardasil-9.

Thus the authors should go the extra mile and show claimed broad-spectrum vaccine efficacy in the challenge model, which is now readily available in many labs worldwide.

Response: We appreciate the reviewer's constructive comments during the review process. We acknowledge that *in vivo* genital HPV PsV challenge model in mouse or rabbits is more sensitive for detecting protective antibody levels, as described in our response to the first round of comments. Due to the unavailability of this model, we claimed the importance of *in vivo* model for our work in the Discussion section, "However, while we have used an established surrogate assay (PBNA) for assessing protection induced by the HPV nine-valent chVLP vaccines, protection should be further tested in an *in vivo* challenge model." (Page 18, line 584-602)